# Structural insights into perilipin 3 membrane association in response to diacylglycerol accumulation

Yong Mi Choi [1,8], Dalila Ajjaji[2,8], Kaelin D. Fleming [3], Peter P. Borbat[4,5], Meredith L. Jenkins[3], Brandon E. Moeller[3], Shaveen Fernando [6], Surita R. Bhatia [6], Jack H. Freed [4,5], John E. Burke [3,7] ✉, Abdou Rachid Thiam [2] ✉ & Michael V. Airola [1] ✉

Lipid droplets (LDs) are dynamic organelles that contain an oil core mainly composed of triglycerides (TAG) that is surrounded by a phospholipid monolayer and LD-associated proteins called perilipins (PLINs). During LD biogenesis, perilipin 3 (PLIN3) is recruited to nascent LDs as they emerge from the endoplasmic reticulum. Here, we analyze how lipid composition affects PLIN3 recruitment to membrane bilayers and LDs, and the structural changes that occur upon membrane binding. We find that the TAG precursors phosphatidic acid and diacylglycerol (DAG) recruit PLIN3 to membrane bilayers and define an expanded Perilipin-ADRP-Tip47 (PAT) domain that preferentially binds DAG-enriched membranes. Membrane binding induces a disorder to order transition of alpha helices within the PAT domain and 11-mer repeats, with intramolecular distance measurements consistent with the expanded PAT domain adopting a folded but dynamic structure upon membrane binding. In cells, PLIN3 is recruited to DAG-enriched ER membranes, and this requires both the PAT domain and 11-mer repeats. This provides molecular details of PLIN3 recruitment to nascent LDs and identifies a function of the PAT domain of PLIN3 in DAG binding.

Lipid droplets (LDs) act as energy reservoirs in cells. They contain a neutral lipid core of mainly triacylglycerols (TAGs) and cholesterol esters with a phospholipid monolayer that surrounds the neutral lipid core. This creates a membrane environment for LDs that is distinct from membrane bilayers and recruits several LD-associated proteins[1–3]. In addition to a major role in energy storage, LDs are also important cellular hubs that traffic proteins and lipids between organelles, regulate ER stress, and contribute to viral infections[4–6].

LDs are formed in the ER where neutral lipids are synthesized[7,8]. Mechanistically, LD formation involves several steps[9] including accumulation of neutral lipids in the outer ER membrane leaflet, neutral lipid nucleation aided by the seipin complex and associated factors (e.g. LDAF1)[10], formation of nascent LDs that bud from the ER[11,12] and LD growth and maturation[13,14]. Several lines of evidence support the idea that the phospholipid composition of the ER membrane is locally edited to promote LD assembly or recruit specific proteins important

[1]Department of Biochemistry and Cell Biology, Stony Brook University, Stony Brook, NY 11794, USA. [2]Laboratoire de Physique de l'École normale supérieure, ENS, Université PSL, CNRS, Sorbonne Université, Université Paris Cité, F-75005 Paris, France. [3]Department of Biochemistry and Microbiology, University of Victoria, Victoria, BC V8N 1A1, Canada. [4]National Biomedical Resource for Advanced Electron Spin Resonance Technology (ACERT), Cornell University, Ithaca, NY 14853, USA. [5]Department of Chemistry and Chemical Biology, Cornell University, Ithaca, NY 14853, USA. [6]Department of Chemistry, Stony Brook University, Stony Brook, NY 11794, USA. [7]Department of Biochemistry and Molecular Biology, The University of British Columbia, Vancouver, British Columbia V6T 1Z3, Canada. [8]These authors contributed equally: Yong Mi Choi, Dalila Ajjaji. ✉e-mail: jeburke@uvic.ca; thiam@ens.fr; michael.airola@stonybrook.edu

for LD formation such as seipin and ORP proteins[7,15–17]. Although there is still uncertainty regarding the phospholipid composition at the initial stages of LD formation, evidence from yeast suggests diacylglycerol (DAG), the direct precursor of TAG, is enriched in discrete ER subdomains where LD biogenesis is initiated[15,18]. DAG can also promote TAG nucleation and impact the architecture of LDs on the ER[15,19].

Perilipins (PLINs) are the major class of proteins that coat the surface of LDs[20–26]. There are five PLINs in humans that bind LDs at various stages of their initiation and maturation[21]. For example, PLIN1 is the major PLIN that binds to mature LDs in adipocytes, while PLIN2 and PLIN5 reside on mature LDs in liver and muscle cells, respectively[27]. In contrast, PLIN3 displays near ubiquitous expression and binds to early LDs as they bud from the ER but is later displaced by other PLINs as LDs grow and mature[10,28]. PLIN3 is stable and not degraded in the absence of LDs. This allows PLIN3 to translocate from the cytoplasm to sites of early LD formation, where PLIN3 is well established to act as a marker for the biogenesis of early LDs across species[10,28,29].

PLINs share a conserved protein domain architecture composed of an N-terminal PAT domain, followed by variable stretches of 11-mer repeats and a C-terminal 4-helix bundle[30,31]. The 11-mer repeats form

amphipathic helixes that are sufficient to recruit PLINs to LDs[30–32], while the 4-helix bundle in some PLINs can also bind LDs[30,33]. The binding of amphipathic helices, such as those found in the 11-mer repeats, to membrane interfaces is greatly influenced by the level or presence of phospholipid packing defects[34,35]. The degree of packing is determined by the type of lipid present (as indicated in the triangle in Fig. 1a). When it comes to the oil-water interface of LDs, the recruitment of amphipathic helices is greater when the level of phospholipid packing is lower[35,36], i.e. when there are more packing defects. Although the PAT domain is the most conserved domain among PLINs, its function is not clear, as it has not been demonstrated to bind to membranes or LDs.

Here, we examine the mechanism of PLIN3 recruitment to membrane bilayers and LDs using a combination of in vitro and cell culture assays, and analyze the structural changes induced by membrane binding using hydrogen-deuterium exchange mass spectrometry (HDX-MS) and pulsed-dipolar electron spin resonance spectroscopy (PD-ESR). We found that *human* PLIN3 is recruited to membrane bilayers enriched in the TAG precursors phosphatidic acid (PA) and DAG. By delineating the roles of the PAT domain and 11-mer repeats, we define an expanded PAT domain that is sufficient for PLIN3 to bind

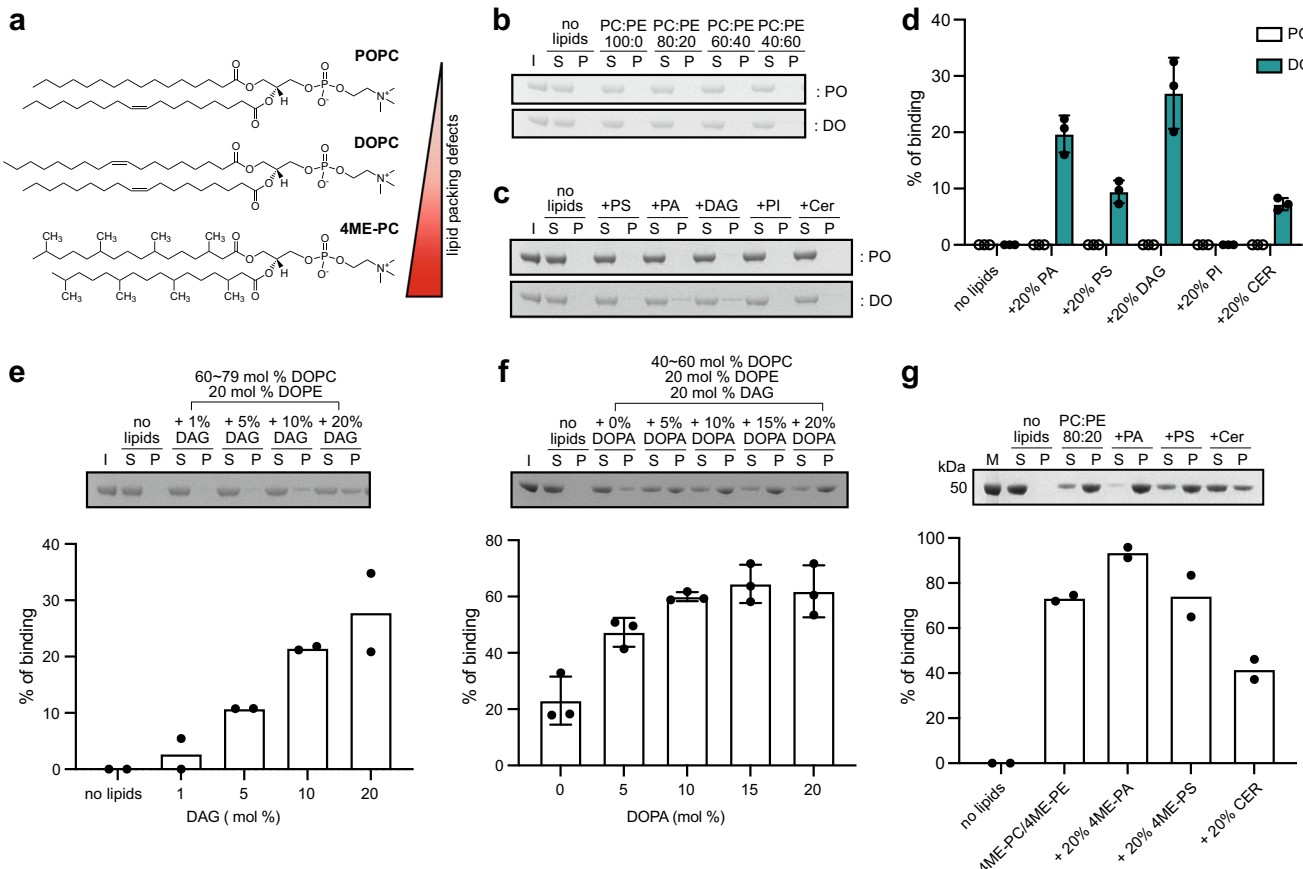

**Fig. 1 | Triglyceride precursors DAG and PA recruit PLIN3 to membrane bilayers. a** Structures of representative phospholipids used in this study. The level of lipid packing defects are depicted as gradient color from red (strong) to pale red (weak). **b** SDS-PAGE analysis of *human* PLIN3 liposome sedimentation assay using either palmitoyl-oleoyl (PO)-liposomes or di-oleoyl (DO)-liposomes with different ratios of PC to PE (n = 3). Lane S (supernatant) and P (pellet) represent unbound and bound *human* PLIN3 to liposomes. Lane I represents input of protein. **c** SDS-PAGE analysis of human PLIN3 recruitment to liposomes enriched with 20 mol% of the additional lipids PS, PA, DAG, PI and ceramide (n = 3). **d** Quantification of PLIN3 recruitment to liposomes in different lipid compositions. Data are presented as mean values +/− SD from three independent experiments (n = 3). **e** SDS-PAGE analysis of *human* PLIN3 recruitment to DO-liposomes using different amounts of

DAG. DOPE concentration was held constant at 20 mol% and DOPC was adjusted between 60 and 79 mol% to maintain total phospholipid concentration. Bar graph shows addition of DAG increases PLIN3 binding (n = 2). **f** SDS-PAGE analysis of *human* PLIN3 recruitment to DO-liposomes enriched with different amount of DOPA. The amounts of DOPE and DAG were kept constant at 20 mol%. DOPC concentration was adjusted between 40 and 60 mol% to maintain total phospholipid concentration. Bar graph shows addition of DOPA further increases PLIN3 binding. Data are presented as mean values +/− SD (n = 3). **g** SDS-PAGE analysis of *human* PLIN3 recruitment to 4ME liposomes enriched with 20 mol% of the additional lipids 4ME-PA, 4ME-PS, ceramide. 4ME-PC and 4ME-PE concentrations ranged between 60–80 mol% and 20 mol%, respectively (n = 2). Lane M represents protein ladder marker. Source data are provided as a Source Data file.

DAG-enriched membranes, while the 11-mer repeats are sufficient to bind LD monolayers. We confirm that DAG enrichment can drive PLIN3 recruitment to ER membranes in cells, and this requires both the PAT domain and 11-mer repeats. Structurally, the PAT domain and 11-mer repeats form inducible alpha helices to drive membrane association and the PAT domain forms a tertiary structure upon membrane binding. Taken together, this study provides molecular insight into how PLIN3 is recruited to early LDs and reveals a function for the PAT domain of PLIN3 in DAG binding.

## Results

### The triglyceride precursors DAG and PA recruit PLIN3 to membrane bilayers

To systematically define how lipid composition affects PLIN3 recruitment to membrane bilayers, we purified recombinant *human* PLIN3 from *Escherichia coli* and used liposome co-sedimentation assays to monitor membrane binding. Liposomes were prepared using multiple freeze/thaw cycles and characterized by dynamic light scattering (DLS) (Supplementary Fig. 1a). Liposomes were made from phospholipids with different acyl-chain combinations comprised of either palmitoyl-oleoyl (PO) or di-oleoyl (DO) phospholipids (Fig. 1a). PO phospholipids have one unsaturated acyl chain and are representative of the typical lipid composition in ER membranes. DO phospholipids contain two unsaturated acyl chains, have increased membrane packing defects[34,37–39], and are enriched in ER membranes after oleate supplementation, which stimulates LD formation[15,40–42]. Initial liposome co-sedimentation experiments varied the ratio of neutral phospholipids phosphatidylcholine (PC) and phosphatidylethanolamine (PE), as PC and PE represent the major lipids on both the cytoplasmic face of the ER and surface of LDs, PE increases both PLIN2 binding to liposomes[43] and PLIN3 insertion into mixed lipid monolayers at phospholipid-oil interfaces[44] and the ratio of PC to PE has previously been shown to regulate protein distribution on LDs[43,45]. Under all PC-to-PE ratios tested, PLIN3 did not bind liposomes comprised solely of PC and PE (Fig. 1b).

We next asked whether the addition of other lipids that are synthesized in the ER could recruit PLIN3 to membranes by generating PC/PE liposomes containing 20 mol% of either phosphatidic acid (PA), phosphatidylserine (PS), diacylglycerol (DAG), phosphatidylinositol (PI) or C18:1 ceramide. Regardless of lipid composition, PLIN3 did not bind to any PO-based liposomes (Fig. 1c, d). However, PLIN3 was recruited to DO-based liposomes enriched in DAG or PA[37,38,46] (Fig. 1c, d). The addition of DOPS and ceramide resulted in a minor increase in PLIN3 binding in DO-based liposomes, while the addition of PI had no effect (Fig. 1c, d).

PLIN3 recruitment to DO-based liposomes depended on the surface concentration of DAG, with 5 mol% DAG able to induce ~10% binding, and 20 mol% DAG inducing ~30% binding (Fig. 1e). The addition of DOPA further increased PLIN3 binding in DAG-containing liposomes, which suggested a synergistic effect of PA and DAG on PLIN3 membrane recruitment (Fig. 1f). Increasing the total liposome concentration caused the majority of PLIN3 to bind membranes, but 5% of PLIN3 remained in the soluble fraction (Supplementary Fig. 2a). DLS characterization indicated DAG enriched liposomes had two populations based on size (Supplementary Fig. 1c, d), which is likely due to the fusogenic properties of DAG[46]. The addition of DOPA or PLIN3 protein decreased or eliminated the population of the larger (>400 nm) liposomes (Supplementary Fig. 1c, d). We concluded that PLIN3 binds liposomes containing DAG and/or PA, which are notably the membrane-lipid precursors for triglyceride synthesis, and PLIN3 can also remodel membranes, consistent with previous studies[29].

### PLIN3 binds to liposomes with membrane packing defects

LDs have increased membrane packing defects in comparison to membrane bilayers[47]. Previously, PLIN4 has been shown to have increased binding to liposomes composed of methyl branched diphytanoyl (4ME) phospholipids that create shallow lipid-packing defects that can be accessed by hydrophobic insertion of peripheral membrane proteins, such as PLINs[31,48]. The packing defects induced by 4ME phospholipids are greater than DO phospholipids because of the wider space between lipids[48]. In comparison to DO and PO liposomes, 4ME phospholipids significantly increased PLIN3 liposome association with ~70% binding observed with a mixture of 4ME-PC and 4ME-PE (Fig. 1g). Consistent with our previous observations, 4ME-PA further increased PLIN3 binding, while 4ME-PS had no effect and C18:1 ceramide decreased binding (Fig. 1g). The effect of DAG was unable to be assessed as the addition of DAG to 4ME-PC/PE disrupted proper liposome formation. Taken together, this suggests PLIN3 membrane binding is also dependent on lipid packing defects, and PA can further enhance PLIN3 membrane binding.

### PLIN3 Binding to Artificial Lipid Droplets

As the major function of PLIN3 is to bind to emerging LDs as they bud from the ER, we next assessed the ability of recombinant PLIN3 to bind artificial lipid droplets (ALDs) in vitro using a flotation assay[10,28,29,49]. ALDs were generated with a triolein-neutral lipid core surrounded by a phospholipid monolayer of DOPC and DOPE in a 1:2.5 molar ratio of phospholipids to TAG. ALDs with 80 mol% DOPC and 20 mol% DOPE showed 50% binding of PLIN3 (Supplementary Fig. 2b). The addition of the anionic phospholipids PI4P decreased PLIN3 binding to ALDs. In contrast, PA, PS, and PI slightly decreased binding, ceramide did not affect binding, and DAG slightly increased PLIN3 binding. However, none of these effects were statistically significant except for the effects of PI4P. The modest impact of PA, PS, PI, DAG, and ceramide, which all can induce negative curvature or charge in membrane bilayers, may be due to the basal occurrence of significant phospholipid packing voids on the ALD surface as compared with bilayers;[36,47] while the strongly charged PI4P would instead mask these and diminish binding[47].

### The PAT domain and 11-mer repeats are disordered in the absence of membranes

We next sought to examine how the structure of PLIN3 changes upon membrane binding, first focusing on the structure of PLIN3 in the absence of membranes. For these experiments, we used hydrogen-deuterium exchange mass spectrometry (HDX-MS), which measures the exchange of amide hydrogens with deuterated solvents. This method acts as a readout for protein conformational dynamics with regions that form secondary structures undergoing slower deuterium exchange than disordered regions, which lack intramolecular hydrogen bonds and secondary structure[50,51]. A brief pulse of deuterated solvent is useful for identifying regions within a protein that lack structure compared to ordered regions[52,53].

We determined the absolute exchange of PLIN3 after a 3 sec pulse of deuterium incorporation at 0 °C (equivalent to ~0.3 sec at 20 °C) using a fully deuterated control. After the deuterium pulse, the first 200 residues of PLIN3 comprising the PAT domain and 11-mer repeats were fully deuterated, indicating these regions are completely disordered in the absence of membranes (Fig. 2a). The 4-helix bundle and α/β domain were largely ordered with comparatively low rates of deuterium incorporation (Fig. 2a), which is consistent with prior structural studies of PLIN3[54,55].

### The PAT domain and 11-mer repeats are the major drivers of PLIN3 membrane association

We next sought to determine any conformational changes that occur during membrane binding using HDX-MS. Having established optimal conditions for PLIN3 membrane binding, we measured the deuterium exchange rate over various time points (3, 30, 300, and 3000 sec) in the absence or presence of 4ME liposomes composed of 60 mol% 4ME-PC, 20 mol% 4ME-PE, and 20 mol% 4ME-PA.

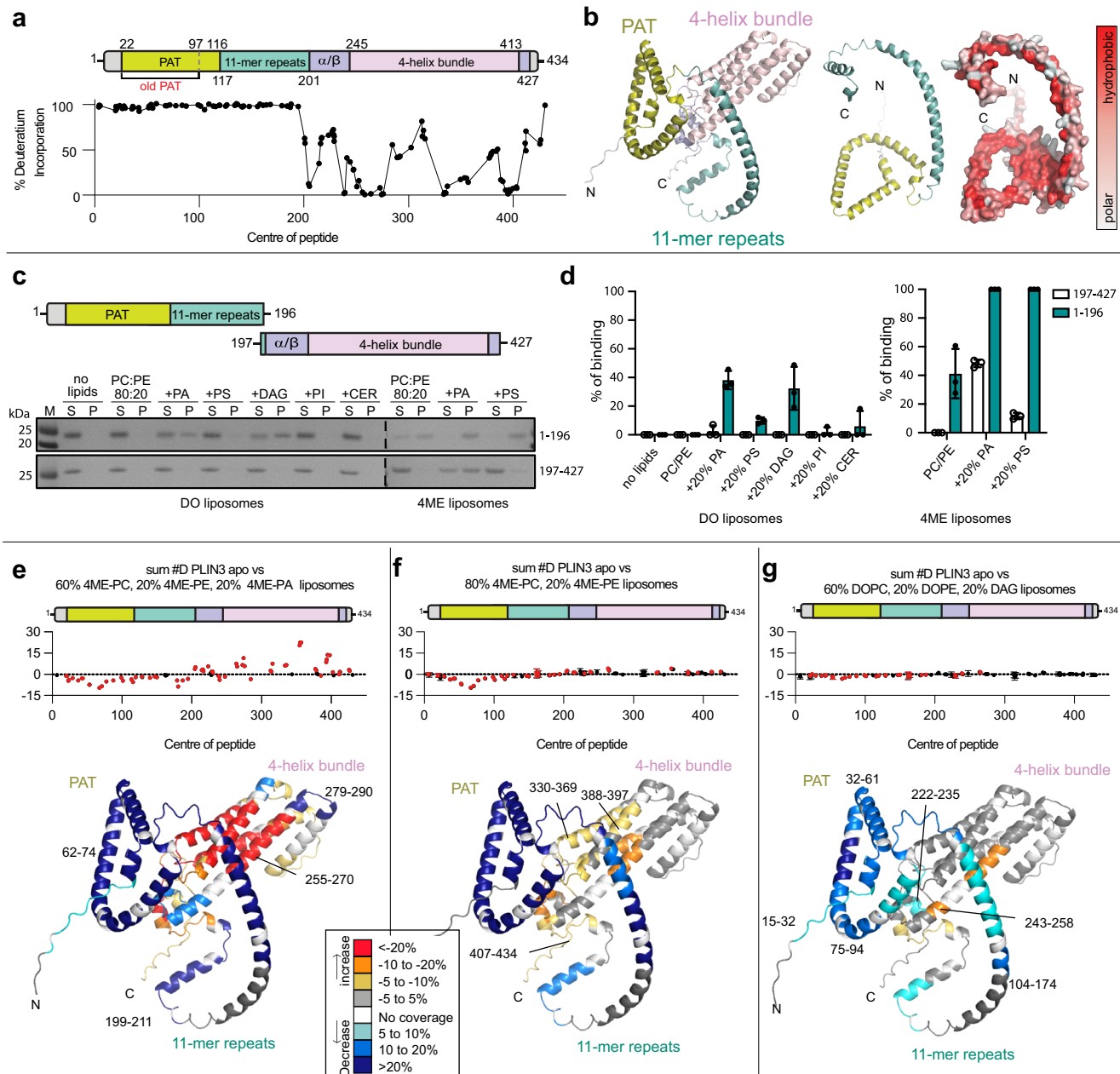

**Fig. 2 | HDX-MS analysis of PLIN3 membrane binding. a** Absolute percentage of deuterium incorporation after 3 sec deuterium exposure of PLIN3 at 1 °C in the absence of liposomes. Each point represents a single peptide, with them being graphed on the x-axis according to their central residue. A domain architecture of *human* PLIN3 was drawn to match the scale of x-axis of HDX-MS. N-terminal PAT domain (mustard) and 11-mer repeats (cyan) have no detectable secondary structure in the absence of liposomes. The α/β domain and 4-helix bundle are represented in purple and pale pink, respectively. **b** A predicted 3D structure of *human* PLIN3 (UniProt: O60664) by AlphaFold (Left). The surface of PAT domain and 11-mer repeats were shown according to the hydrophobicity (Right). The level of hydrophobicity is presented as a color gradient from red (hydrophobic) to gray (polar). **c** SDS-PAGE and **d** quantitative analysis of PAT/11-mer repeats and 4-helix bundle liposome binding to DO and 4ME-liposomes. Data are presented as mean values +/− SD from three independent experiments (*n* = 3). Lane M represents protein ladder. **e**–**g** Quantitative analysis of deuterium exchange differences of human PLIN3 in the presence of liposomes. The sum of the difference in the # of incorporated deuterons is shown for the absence and the presence of liposomes over all timepoints. The N-terminus composed of PAT/11-mer repeats were significantly protected from deuterium exchange (defined as > 5% change in exchange, > 0.4 Da mass difference in exchange, a *p*-value <0.01 using a two-tailed Student's t-test). Each point represents an individual peptide, with those colored in red having a significant difference, with error bars showing standard deviation (*n* = 3). Liposomes were generated with **e** 60 mol% 4ME-PC, 20 mol% 4ME-PE and 20 mol% 4ME-PA, **f** 80 mol% 4ME-PC, 20 mol% 4ME-PE or **g** 60 mol% DOPC, 20 mol% DOPE and 20 mol% DAG. A map of deuterium exchange rate according to all peptides throughout the entire PLIN3 was generated based on the AlphaFold predicted PLIN3 structure and color-coded according to the legend. Source data are provided as a Source Data file.

In the presence of membranes, striking differences in deuterium exchange were observed throughout the PLIN3 sequence (Fig. 2e, Supplementary Table 1). The most notable differences were large decreases in deuterium exchange in the PAT domain and 11-mer repeat regions. In comparison to previous results that demonstrated the 11-mer repeats are sufficient for the lipid droplet association[30,31], this suggests that the PAT domain and 11-mer repeats are both major contributors to membrane binding. Consistently, we found that a

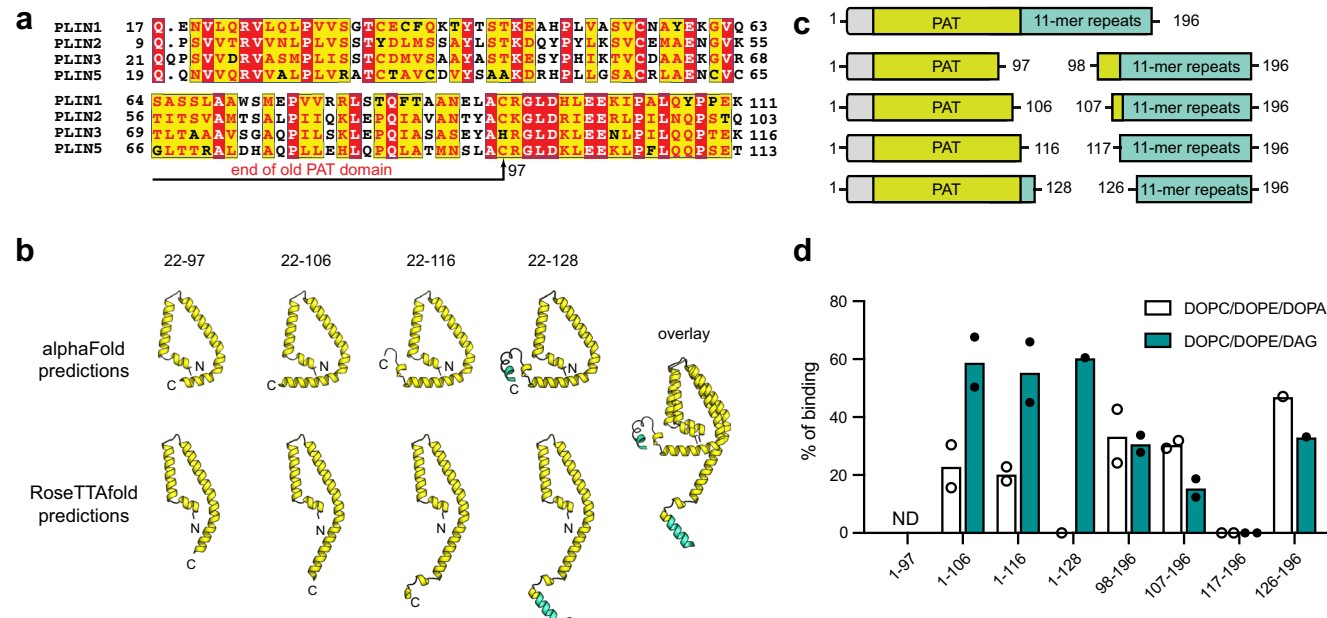

**Fig. 3 | An expanded PAT domain binds DAG-enriched membranes. a** Multiple sequence alignment of the N-terminus of *human* PLIN1 (UniProt: O60240), PLIN2 UniProt: Q99541), PLIN3 (UniProt: O60664), and PLIN5 (UniProt: Q00G26). The previously suggested PAT domain (residues 1–97) is indicated with black arrow. **b** Putative PAT domain triangular structures predicted by both AlphaFold and RoseTTAFold. **c** Schematic of PAT domain constructs and the counterpart 11-mer repeats constructs. **d** Quantitative analysis of liposome recruitment for various PAT domain and 11-mer repeats constructs. Data are presented as mean values from two independent experiments ($n = 2$) for constructs 1–106, 1–116, 98–196, 107–196, and 117–196 and one independent experiment ($n = 1$) for constructs 1–128 and 126–196. The 1–97 construct was not determined (ND) due to an inability to express. Source data are provided as a Source Data file.

purified PAT/11-mer repeats fragment displayed similar membrane recruitment as full-length PLIN3 to DO liposomes enriched in DAG and PA, and 4ME liposomes (Fig. 2c, d).

The HDX-MS results suggested that membrane binding induces the formation of a secondary structure in the PAT domain and 11-mer repeats. This is consistent with both AlphaFold and RoseTTAFold models that predict the PAT domain and 11-mer repeats form amphipathic alpha helices, with the PAT domain adopting a triangular globular structure and the 11-mer repeats forming a series of extended alpha helices (Figs. 2b, 3b). Based on these results we concluded that the PAT domain and 11-mer repeats of PLIN3 are intrinsically disordered in the absence of membranes, with inducible amphipathic alpha helices being stabilized upon membrane binding.

Notably, many of the peptides with the PAT domain and 11-mer repeats displayed a bimodal distribution of deuterium exchange upon membrane binding (Supplementary Fig. 3a, b). This differs from a canonical distribution where the degree of deuteration is gaussian in individual peptides. The observed bimodal distributions are consistent with either EX1 exchange[56], or there being two distinct protein conformations in the sample. Depending on the membrane residency of PLIN3, bimodal distributions can be consistent with observations of the free and membrane-bound states. Given our previous observation that the PAT and 11-mer repeats are intrinsically disordered in the absence of membranes, the simplest interpretation is that the PAT domain and 11-mer repeats cycle between two states: 1) a membrane-bound alpha helical state on membranes that are strongly protected from H-D exchange and 2) an intrinsically disordered solution state that rapidly exchanges with deuterated solvent upon dissociation from membranes and leads to rapid H-D exchange. Therefore, the half-life of bimodal exchange can act as a surrogate for membrane residency time for disordered regions of the protein.

The protection from deuterium exchange and the non-gaussian bimodal distribution for most peptides in the PAT domain persisted over a longer time course (~300 sec) in comparison to peptides within the 11-mer repeat regions (~30 sec) (Supplementary Fig. 4a). This trend of longer HDX protection correlated with the degree of sequence conservation among PLINs (Fig. 3a). Taken together this supports a role for the PAT domain in membrane binding and suggests a similar role for the highly conserved PAT domain in other PLINs.

In contrast to the N-terminal PAT domain and 11-mer repeats, the C-terminal 4-helix bundle displayed a large increase in deuterium exchange (Fig. 2e, Supplementary Fig. 4a). The most apparent increase in deuterium exchange was in the middle of the 4-helix bundle. We hypothesized this was due to the 4-helix bundle unfolding upon membrane binding, with this possibly contributing to membrane recruitment. To test this, we examined the ability of a purified PLIN3 4-helix bundle fragment (residues 197–427) to bind membranes. Consistent with our hypotheses, the isolated 4-helix bundle bound to the 4ME-PC/PE/PA liposomes that were used in the HDX-MS experiment (Fig. 2c, d). However, the 4-helix bundle did not bind to 4ME-liposomes lacking PA, or to any DO-based liposomes even when PA and DAG were present (Fig. 2c, d). We concluded that the 4-helix bundle of PLIN3 can unfold and bind membranes, but only under very specific conditions such as the presence of accumulated PA on the membrane having packing defects induced by 4ME-PC/PE. This conclusion is supported by the PA accumulation at the nascent LD formation site where lipid packing defects occur[17].

To understand how lipid composition influences the structure of PLIN3, we again applied HDX-MS, using variable liposome compositions of 4ME-PC/PE and DOPC/PE/DAG (Fig. 2f, g). In line with of previous HDX-MS and liposome sedimentation experiments, we observed protection from H-D exchange in the PAT domain and 11-mer repeats using 4ME-PC/PE and DOPC/PE/DAG liposomes, with the magnitude and duration of H-D exchange protection in these regions correlating with the observed fraction of PLIN3 bound in liposome sedimentation assays (Supplementary Fig. 4b, Fig. 1d, g).

In contrast, with the 4ME-PC/PE and DOPC/PE/DAG lipid compositions, the 4-helix bundle did not display large increases in deuterium

exchange indicating that 4-helix bundle is not involved in membrane binding in the absence of PA. Taken together, these results confirm that the PAT domain and 11-mer repeats are the major lipid interacting regions of PLIN3, and that the 4-helix bundle forms a stable tertiary structure that in general does not contribute to membrane association, unless PA is present in membrane areas with shallow lipid packing defects. This condition is particularly fulfilled at nascent LD formation sites.

## An expanded PAT domain binds DAG-enriched membranes

Our results suggested a role for the PAT domain in PLIN3 membrane association. However, several studies[30,31] have previously established the 11-mer repeats of PLINs are sufficient for membrane binding, and to our knowledge, the PAT domain alone has not been demonstrated to bind membranes. Complicating matters, the precise boundaries of the PAT domain have been difficult to establish. Previous studies have suggested the PAT domain consists of the N-terminal 97 residues in PLIN3[30], which are highly conserved with other PLINs (Fig. 3a). However, high sequence conservation between PLINs continues beyond residue 97 and ends at residue 115 in PLIN3 (Fig. 3a). This suggests the PAT domain may be larger than previously expected. Consistent with the sequence conservation, both Alphafold[57] and RoseTTA fold[58] predict a triangular alpha-helical tertiary structure for the PAT domain that encompasses residues 22-116 (Fig. 3b).

To investigate the PAT domain boundaries and membrane interaction capabilities, we purified several C-terminal extended PAT domain constructs and the corresponding 11-mer repeats counterparts (Fig. 3c) and assessed the ability of these fragments to bind DO liposomes containing either 20 mol% PA or 20 mol% DAG. As an important note, we were unable to purify the previously suggested PAT domain construct (residues 1–97), as this construct was not stably expressed in *Escherichia coli*. This is consistent with the poor expression of the PAT domain of PLIN1 in yeast[30].

In general, fragments encompassing the PAT domain bound strongly to liposomes in the presence of DAG, with modest liposome binding in the presence of PA (Fig. 3d). In contrast, the 11-mer repeats bound to liposomes containing either PA or DAG (Fig. 3d). Taken together, the data suggests that the PAT region does form a domain that spans residues 22–116 in PLIN3. Notably, this expanded PAT domain is capable of membrane binding and displays a preference for binding DAG enriched membranes, while the 11-mer repeats of PLIN3 do not.

## Conformational rearrangements of the PAT domain upon membrane binding

We first sought to confirm the effects of liposome binding on the secondary structure of PLIN3 by circular dichroism (CD) (Supplementary Fig. 5). Liposomes were prepared with 4ME-PC/PE/PA, which recruited all the PLIN3 constructs (full length, PAT, 11-mer repeats, PAT/11-mer repeats, 4-helix bundle) in the previous liposome co-sedimentation experiments. For full-length PLIN3, the presence of liposomes increased overall helicity as observed by an increased negative peak around 222 nm in the CD spectra. In comparison, the CD spectra of the 4-helix bundle were largely unaffected by the presence of liposomes and were consistent with a stable alpha-helical structure. In contrast, the PAT/11-mer repeats adopted a mostly random coil structure in solution with a negative peak around 200 nm, and a shift to alpha helices in the presence of liposomes as indicated by a large negative peak at 222 nm. Liposomes induced similar changes in the CD spectra for both the PAT domain and 11-mer repeats alone. Taken together, this confirms that the increase in helicity observed in full-length PLIN3 by membranes was due to the PAT/11-mer repeats undergoing a disorder/alpha helical transition. This is in line with our HDX-MS results and prior studies of PLIN2 and PLIN3 fragments[55,59].

Next, to investigate if membranes induced conformational changes in the PAT domain, we applied the pulsed-dipolar spectroscopy (PDS) technique of double electron-electron resonance (DEER)[60,61] to full length PLIN3 in solution and bound to liposomes. PDS is a collection of several, based on recording electron spin-echo (ESE), pulse ESR techniques[62-66] routinely applied to characterize protein conformations by providing accurate constraints in the distance range of ~10–90 Å. The amplitude of detected ESE depends on dipolar coupling between the spins of unpaired electrons in nitroxide spin labels covalently bound to engineered cysteine residues. Stepping out pulse separation in the sequence produces the time domain ESE envelope, from which the distances could be reconstructed[62-65].

Two sets of residues (25 and 96; 37 and 114) were selected for spin labeling, which were respectively 71 and 77 residues apart in the primary sequence but predicted to be in close proximity by AlphaFold and RoseTTAFold (Fig. 4a–c). Comparison of the DEER decoherence (decay) times indicated that the PAT domain is less structured and/or intrinsically disordered in solution versus when bound to liposomes, which is consistent with our HDX-MS and CD analysis (Fig. 4d). One can estimate, based on ~20% extent of decay in solution compared to liposomes that the decoherence time is of the order of 20 μs, which corresponds to nearly a 100 Å separation expected for a random polypeptide chain with stiffness[67]. In contrast, with liposomes, the decoherence times are fast enough to decay to the background well within the evolution time of 1.2 μs used in the liposome DEER measurements. The decays do not show oscillations, as is frequently observed for spin pairs with narrow distance distributions[68-71]. As a control to rule out lateral aggregation[69], we attached a single spin-label to residue 37C and did not observe a dipolar evolution (DEER shape) that could indicate a pair with a shorter than ~80 Å separation based on only slightly concave shape of the signal (Fig. 4e). The DEER data thus suggest that conspicuous dipolar signals in the doubly labeled proteins result from intramolecular interactions, rather than being caused by intermolecular interactions on membranes.

Distance distributions could be obtained in the presence of liposomes. The results showed semi-broad distance spreads of ~20-40 Å for the 25C/96C pair and ~20-35 Å for the 37C/114C pair (Fig. 4f, g). These distances are comparable but do not exactly match the simulated distances using the AlphaFold and RoseTTAFold structure predictions that have only modest confidence levels (Fig. 4a)[57,58]. For comparison, an extended alpha helix would result in distances of ~100 Å for these two sites. We also checked the spectral shape by recording field-swept echo with pulse separation of 250 ns and did not notice any conspicuous broadening that could indicate a shorter range of distances (<15 Å). We do however see from continuous wave (CW) ESR of 37C/114C (Supplementary Fig. 6) that there might be a sizeable fraction of spins in the 15-20 Å range whose contribution to the distance distribution will be significantly attenuated since DEER has low sensitivity to distances in this range. The conformations with this distance range correlate well with AlphaFold predictions. The spread of the P(r) to longer distances could originate from the mobility of the C-terminal helix where residue 114C is located (Fig. 4c). Taken together, we concluded that the PAT domain does form a folded domain when bound to membranes and this domain is likely mobile with a structure similar but not identical to the AlphaFold and RoseTTAFold predictions.

## DAG recruits PLIN3 to membrane bilayers and droplets in vitro

To verify a role for the PAT domain in DAG binding, we conducted an independent set of experiments to test if purified PLIN3 with GFP fused to the N-terminus bound DAG enriched membranes using droplet embedded vesicles (DEVs)[7,35,45,72]. DEVs have emerged as a powerful in vitro system to model emerging LDs that bud from membrane bilayers. A major advantage of DEVs is the ability to not only monitor

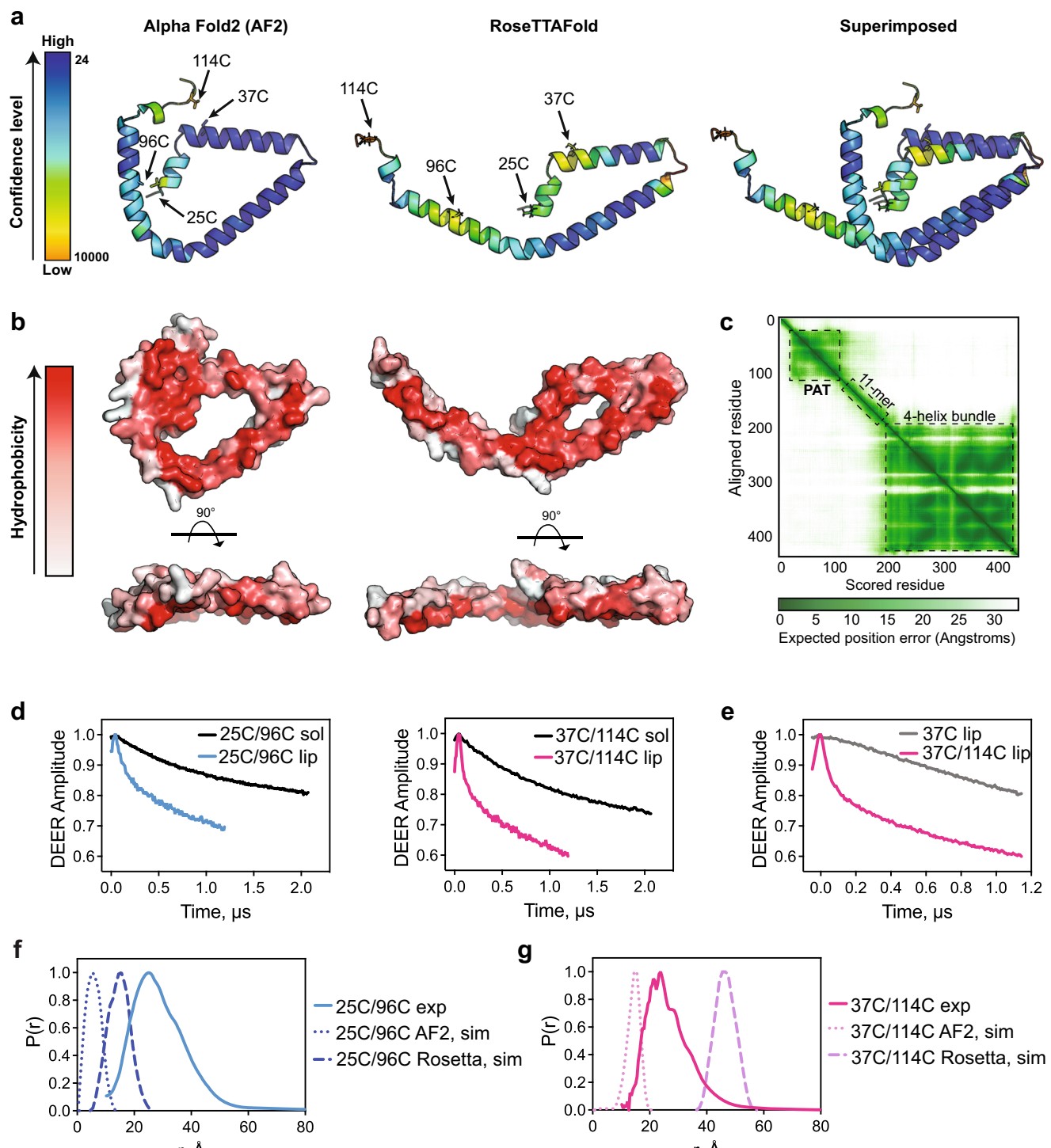

**Fig. 4 | Membrane-induced conformational rearrangements are consistent with a PAT domain tertiary structure. a** Cartoon representation of the PAT domain structure predicted by AlphaFold2 and RoseTTAFold with relative confidence levels shown. The positions of the spin-labeled residues are indicated with arrows. The relative confidence level was colored in gradient from orange (Low) to deep navy (High). **b** The hydrophobic surface of the predicted PAT domain is depicted. The level of hydrophobicity is presented as color gradient from red (hydrophobic) to gray (polar). The hydrophobic face of the helices is facing towards the reader. **c** The Predicted Aligned Error (PAE) values from AlphaFold2 for full-length PLIN3 were plotted by ChimeraX and shown as a 2D plot (bottom, right panel). **d** Time domain signals from DEER spectroscopy of double-labeled 25C/96C (left) and 37C/114C (right) in solution (sol) and on liposomes (lip). **e** Timedomain signals from DEER spectroscopy of double-labeled 37C/114C and single-labeled 37C proteins on liposomes. **f, g** Distance distributions from DEER spectroscopy of (**e**) 25C/96C and (**f**) 37C/114C determined experimentally (exp) and from MtsslWizard simulations using the AlphaFold2 (AF2, sim) and RoseTTAFold (Rosetta, sim) predicted structures. Source data are provided as a Source Data file.

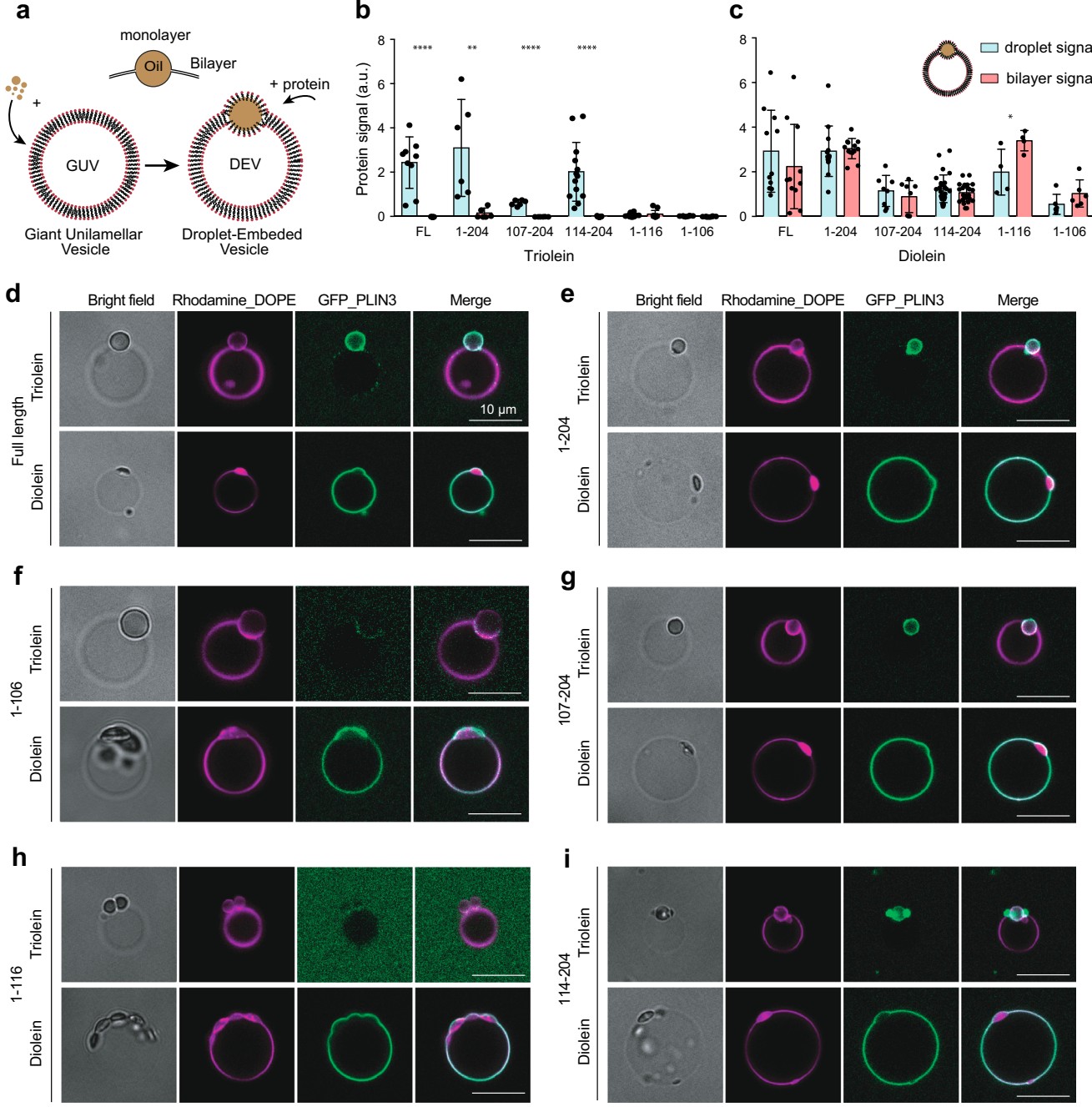

**Fig. 5 | DAG recruits PLIN3 to membrane bilayers and LD droplets in vitro. a** A diagram of generation of DEVs by mixing oils and giant unilamellar vesicles (GUVs). **b, c** Quantification of the recruitment of GFP tagged PLIN3 constructs to droplet embedded vesicles (DEVs) containing **b** triolein or **c** Diolein. Statistical analysis was done with student t-test. ****, $P < 0.001$; **, $P < 0.1$. Three different experiments were performed and 10 to 15 DEVs were quantified for each experiment. Mean and SD is shown for each bar graph. Source data are provided as a Source Data file.

**d–i** Fluorescent microscopic images of N-terminal GFP tagged full length and PLIN3 constructs (green) comprised of PAT domain and 11-mer repeats to (DEVs) (magenta) that are generated with fluorescent-labeled phospholipids and oils. **f, h** Recruitment of PAT domain constructs. Almost no recruitment of PAT domain constructs was observed in the presence of Triolein. **g, i** Recruitment of 11-mer repeats constructs.

membrane association but to also examine the protein distribution and preference for droplet monolayers versus membrane bilayers.

DEVs were generated by adding the neutral lipids TAG or DAG to giant unilamellar vesicles (GUVs) of DOPC that were marked with fluorescent phospholipids (Fig. 5a). As observed previously[45,73], full-length PLIN3 was strongly recruited to the monolayer surface of triolein droplets with almost no bilayer signal (Fig. 5b, d). The addition of diolein (DO-DAG) generated small droplet buds, and full-length PLIN3 was recruited equally to the droplet and bilayer surface (Fig. 5c, d).

Residues 1-204 of PLIN3 that contained both the PAT domain and 11-mer repeats behaved nearly identical to full length PLIN3 in both TAG and DAG containing DEVs (Fig. 5b, c, and e).

Next, we tested fragments of the PAT domain and 11-mer repeats using both triolein and diolein DEVs. The 11-mer repeats preferentially bound the droplet surface of triolein DEVs with the magnitude of signal dependent on the length of the 11-mer repeats (Fig. 5b, g, and i). In contrast, PAT domain fragments did not bind to the droplet or bilayer surface of triolein DEVs (Fig. 5b, f, and h). The PAT domain was

recruited to diolein DEVs, and the largest fragment (residues 1–116) bound to both the droplet and bilayer surface with similar magnitudes to full-length PLIN3 (Fig. 5c, f, and h). The 11-mer repeats are also bound to both the droplet and bilayer surface of diolein DEVs, but to a lesser extent (Figs. 5c, g, and i). These results are consistent with our previous hypothesis that the PAT domain is larger than previously expected and that this functional PAT domain binds DAG-enriched membranes, while the 11-mer repeats can also bind DAG-enriched membranes but display a preference for TAG-containing droplets over membrane bilayers.

### DAG accumulation is sufficient to recruit PLIN3 to the ER in cells

DAG has previously been proposed to recruit PLIN3 to the ER in cells by blocking its hydrolysis or acylation or by the exogenous addition of DAG[74]. This is consistent with the current model for LD formation, where DAG accumulates at the site of TAG nucleation on the ER membrane[18,19,75], the local high concentration of neutral lipids promotes LD nucleation through seipin[15], and cytoplasmic PLIN3 marks these sites[10,49]. We sought to test whether DAG accumulation was sufficient for PLIN3 recruitment using an independent system and also assess what fragments of PLIN3 were necessary and sufficient for ER recruitment.

To best visualize PLIN3 recruitment to ER membranes, we generated intracellular giant ER vesicles (GERVs) by submitting cells to hypotonic medium[72,76,77] (Fig. 6a). After exchange to hypotonic medium, cells were pretreated with DMSO or the DGAT inhibitors (DGATi), followed by oleic acid to induce TAG or DAG synthesis[77] (Fig. 6a). Confocal microscopy was used to visualize cells prior and after exchange to hypotonic medium, and after oleic acid treatment. Imaging was done in the following minutes after the treatments.

As expected, the majority of subcellular localization of GFP-tagged PLIN3 was cytoplasmic in both normal and hypotonic media without the addition of oleic acid (Fig. 6b), and co-localized with LDs (Fig. 6d). Under conditions of DGAT inhibition, PLIN3 co-localized to the outer periphery of GERVs (Fig. 6b, e, f), which suggests DO-DAG accumulation is sufficient to recruit PLIN3 to ER membrane bilayers.

In line with our previous results, a construct containing both the PAT domain and 11-mer repeats showed a similar subcellular localization as full-length PLIN3 under all conditions (Fig. 6c–f). This suggests the PAT domain and 11-mer repeats are sufficient for PLIN3 recruitment to both DAG-enriched membrane bilayers and TAG-containing LDs, which might also contain DAG whose concentration could increase the protein binding level. Interestingly, constructs containing only the PAT domain (residues 1–116) or only the 11-mer repeats (residues 114–204), which bound to DEVs in a DAG-dependent manner, remained cytoplasmic under all conditions (Fig. 6d, e, Supplementary Fig. 7). Taken together, we concluded that PLIN3 is capable of binding both DAG enriched ER membranes and early LDs, and that the PAT domain and 11-mer repeats are both necessary and synergize to perform these functions in cells.

## Discussion

Here we find that PLIN3 not only binds LDs, but also membrane bilayers enriched in DAG and PA. These results are in line with other studies that revealed PLIN3 binding to DAG[74,78,79]. In our hands, DAG binding was observed with multiple in vitro systems (e.g. liposomes, DEVs) and DAG was also sufficient for PLIN3 recruitment to the ER in cells. Cellular recruitment to membranes by DAG is consistent with a seminal study that used inhibitors to block DAG hydrolysis or acylation and exogenous DAG to promote ER recruitment of PLIN3[74]. Independent experiments in yeast have also demonstrated that membrane-anchored PLIN3 is sufficient to bind to DAG-enriched subdomains in the ER[78,79]. Here we show that oleate addition in combination with DGAT inhibition is sufficient for the subcellular redistribution of PLIN3 to the ER, which mimics the normal LD biogenesis pathway.

There are several mechanistic implications of PLIN3 recruitment to DAG-enriched membranes. First, this implies that PLIN3 is recruited to sites of LD formation, not only through TAG generation but also at the initial stages when DAG begins to accumulate. This raises a likely possibility that PLIN3 may play an active role in the early stages of LD formation by stabilizing DAG-enriched regions on ER[78], and defining sites of LD formation, before or in concomitant with seipin. Once PLIN3 binds to accumulated DAG, it might coat the curved surface of a growing DAG/TAG lens and regulate LD budding, in conjunction with seipin.

PA is another TAG precursor that is present at sites where LDs originate[17] and can bind to seipin[16]. In addition to DAG, PA was found to have a significant impact on the recruitment of PLIN3 to membranes in vitro. Thus, it may increase the translocation of PLIN3 to early LD formation sites. These two TAG precursors appear to provide specificity for the association of PLIN3 with membranes. From a curvature standpoint, DAG has a more negative curvature compared to PA and PE, whereas PE has a more negative curvature than PA[80]. This suggests that membrane curvature cannot solely account for the major role of PA in PLIN3 membrane binding specificity. However, from a surface charge perspective, PA and PE together may act synergistically by increasing the charge of PA on the membrane[46,81,82]. In this scenario, PLIN3 recruitment to LD nucleation sites could be enhanced by specific recognition of PA, potentially through the 4-helix bundle. Therefore, PLIN3 membrane association may not only be determined by membrane packing defects but could also involve selective physical interactions between PLIN3 and DAG or PA. This idea is also supported by a previous study that found that the LD binding properties of PLINs are sensitive to the polar residue composition of their amphipathic helices[83].

In this study, we attempted to clarify the function and boundaries of the PAT domain. Our HDX-MS results using full-length PLIN3 clearly implicate both the PAT domain and 11-mer repeats in membrane binding. In addition, we were able to define a functional PAT domain that encompasses all of the conserved residues within PLINs and is longer than previously suggested. This expanded PLIN3 PAT domain is sufficient to bind DAG-enriched membranes, but not LDs. In contrast, the 11-mer repeats display some affinity for DAG-enriched membranes and are necessary to bind LD monolayers. Our overall conclusion is that the PAT domain and 11-mer repeats serve synergistic functions, as the individual regions are necessary for both LD and DAG recruitment in cells.

The PAT domain is predicted to adopt a triangular tertiary structure by both AlphaFold and RoseTTAFold. The DEER distance measurements and CW ESR data are not identical to these predictions but do indicate that when bound to membranes the PAT domain adopts a folded domain. This conclusion is supported by our HDX-MS results that found the peptides within the PAT domain display longer protection times from H-D exchange, which could be due to either a tertiary structure more resistant to unfolding or a longer membrane residency time. We note that the membrane-bound PAT domain structure is likely dynamic and additional distance measurements at distinct sites are needed to verify the accuracy of the predicted triangular structures.

Our finding that the PAT domain of PLIN3 is sufficient to bind DAG represents a discovery of a functional role for the PAT domain in a PLIN. This suggests that the highly conserved PAT domain in other PLINs may serve similar or related functions. For example, could the PAT domain be a general sensor for DAG in membrane bilayers? Or could the PAT domain of other PLINs potentially bind to other neutral lipids (e.g. CEs, TAGs, retinol esters)? Given the previously observed differing neutral lipid preference of PLINs[20] and the involvement of the PAT domain in lipid/membrane binding, this now raises the questions if this is due to the PAT domain preference, the 11-mer repeat preference, or the synergistic action of the combined PAT/11-mer repeat units.

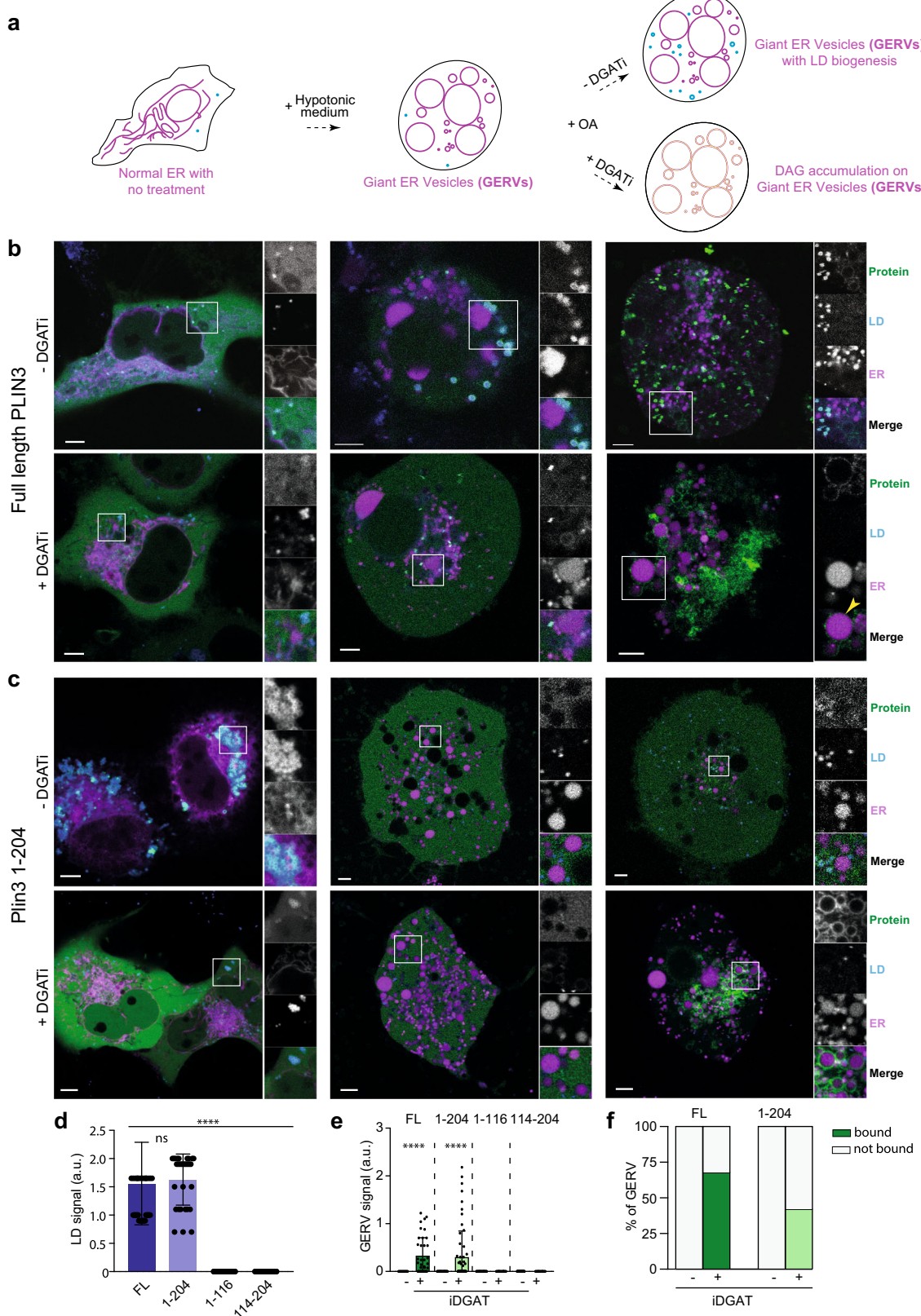

Lastly, our results are consistent with previous studies that found the PLIN3 4-helix bundle does not bind LDs, but other PLIN 4-helix bundles can bind LDs[33]. While the domain boundaries of the 4-helix bundles seem reasonable given structure of mouse PLIN3[54], the variable membrane-binding affinities we observed for the PAT domain and 11-mer repeats does raise the question of the stability of 4-helix bundle,

and consequently the ability to unfold and bind membranes or LDs in other PLINs may depend on the domain boundaries of specific constructs. Thus, we suggest it is reasonable to revisit the ability of the 4-helix bundles from other PLINs using constructs that can be stably purified in vitro to rule out potential artifacts from the use of unstable 4-helix bundles.

**Fig. 6 | Both PAT domain and 11-mer repeats are necessary for PLIN3 recruitment to DAG-enriched ER membranes in Cos7 cells. a** A diagram of treatment of Giant ER Vesicle formation and treatment of oleic acid and DGATi to induce DAG accumulation on the ER in Cos7 cells. **b**, **c** Subcellular localization of GFP tagged full-length PLIN3 and PAT/11-mer repeats (1-204) were visualized in green in Cos7 cells under fluorescent microscope ZEISS LSM800 Airyscan. ER was visualized with ER-specific marker RFP-KDEL in magenta. Lipid droplets were labeled with LipidTox Deepred to stain neutral lipids in cyan. After being treated with a hypotonic medium, cells were supplemented with oleate in the presence or absence of DGAT1/2 inhibitors. Each experiment was performed more than 3 times. **d** Quantification of the amount of various PLIN3 constructs on LDs. Data are presented as mean values +/− SD from five independent experiments (n = 5). Statistical analysis was done with a student t-test. ****, P < 0.001. **e** Quantification of the amount of various PLIN3 constructs on GERV after oleate treatment in the presence and the absence of DGAT1/2 inhibitors. Data are presented as mean values +/− SD from four independent experiments (n = 4). Statistical analysis was done with a student t-test. ****, P < 0.001. **f** Percentage of GERV that are covered with full-length PLIN3 or PAT/11-mer repeats construct (1-204) after oleate treatment in the presence and the absence of DGAT1/2 inhibitors. Source data are provided as a Source Data file.

## Methods

### Protein expression and purification
The genes encoding PLIN3 constructs were codon-optimized for expression in Escherichia coli and cloned into pTHT, which is a derivative of pET28 that contains a TEV cleavable N-terminal 6x His tag. For GFP tagged PLIN3 constructs, monomeric superfolder GFP (msfGFP) was inserted between the 6x His tag and the N-terminus of PLIN3. PLIN3 plasmids were transformed into BL21(DE3) RIPL cells and protein expression was induced with 1 mM isopropyl β-D-1-thiogalactopyranoside (IPTG) at 37 °C for 3 h. Cells were harvested by centrifugation at 3320 x g for 20 mins and stored at −80 °C until use. Cell pellets were resuspended with buffer A containing 500 mM NaCl, 20 mM Tris-HCl pH 7.5, 5% glycerol, 5 mM 2-mercaptoethanol (BME), and lysed by sonication. Cell lysates were centrifuged at 81,770 x g for 1 h at 4 °C and the resulting supernatant was incubated with 5 mL of Ni-NTA resin for 2 h at 4 °C prior to loading onto a gravity column. The Ni-NTA resin was washed with buffer B containing 500 mM NaCl, 20 mM Tris-HCl pH 7.5, 5% glycerol, 5 mM BME, and 60 mM imidazole. Human PLIN3 proteins were eluted with buffer C containing 500 mM NaCl, 20 mM Tris-HCl pH 7.5, 5% glycerol, 5 mM BME, and 300 mM imidazole. Eluted protein was applied to a HiLoad 26/600 Superdex 200 pg column (GE life sciences) equilibrated with 150 mM NaCl, 20 mM Tris-HCl pH 7.5, 5% glycerol, and 5 mM BME. The average yield of full-length PLIN3, PAT/11-mer repeats, and 4-helix bundle throughout expression and purification were between 5–10 mg from a 1 L culture of Escherichia coli. The yield of PAT or 11-mer repeats constructs or GFP-tagged PLIN3 fragments was ~1 mg/L. Protein was analyzed by SDS-PAGE, concentrated using a centrifugal tube (10K MWCO, Pall Corporation), aliquoted, and stored at −80 °C.

### Lipids
The following lipids were purchased from Avanti Polar Lipids: 1,2-dioleoyl-sn-glycero-3-phosphocholine (DOPC, catalog #850375), 1,2-dioleoyl-sn-glycero-3-phosphoethanolamine (DOPE, catalog #850725), 1,2-dioleoyl-sn-glycero-3-phosphate (DOPA, catalog #840875), 1,2-dioleoyl-sn-glycero-3-phospho-L-serine (DOPS, catalog #840035), 1-palmitoyl-2-oleoyl-glycero-3-phosphocholine (POPC, catalog #850457), 1-palmitoyl-2-oleoyl-sn-glycero-3-phosphoethanolamine (POPE, catalog #850757), 1-palmitoyl-2-oleoyl-sn-glycero-3-phosphate (POPA, catalog #840857), 1-palmitoyl-2-oleoyl-sn-glycero-3-phospho-L-serine (POPS, catalog #840034), 1,2-diphytanoyl-sn-glycero-3-phosphocholine (4ME-PC, catalog #850356), 1,2-diphytanoyl-sn-glycero-3-phosphoethanolamine (4ME-PE, catalog #850402), 1,2-diphytanoyl-sn-glycero-3-phosphate (4ME-PA, catalog #850406), 1,2-diphytanoyl-sn-glycero-3-phospho-L-serine (4ME-PS, catalog #850408), 1-palmitoyl-2-oleoyl-sn-glycerol (DAG, catalog #800815), L-α-phosphatidylinositol (Soy PI, catalog #84044), L-α-phosphatidylinositol-4-phosphate (Brain, Porcine (PI(4)P, catalog #840045), N-stearoyl-D-erythro-sphingosine (C18:1 ceramide, catalog #860518). 1,2,3-Trioleoyl Glycerol was purchased from Cayman Chemicals (TAG, catalog #26950).

### Liposome co-sedimentation assay
To prepare liposomes, lipids were dried under a nitrogen gas stream and dissolved with buffer containing 50 mM NaCl and 50 mM HEPES pH 8.0 with a total lipid concentration of 2 mM, unless otherwise noted. Liposomes were generated by repeated cycles of flash freezing in liquid nitrogen, thawing in 37 °C water bath, and vortexing[84]. Liposomes were characterized using dynamic light scattering and showed uniform distribution for most lipid compositions. Lamellarity analysis of liposomes was not carried out, but liposomes likely contained a mixture of multi-lamellar vesicles and unilamellar vesicles. For liposome co-sedimentation assays, 20 μL of 12 μM purified protein was incubated with 40 μL of 2 mM liposomes for 40 mins at 23 °C. Protein fractions bound to liposomes were isolated by centrifugation at 100,000 x g for 70 mins at 4 °C and analyzed by SDS-PAGE. All the scanned gel images from the liposome co-sedimentation assay are available in the Source Data file.

### Circular dichroism
Circular dichroism (CD) spectra of purified PLIN3 full length and its constructs were measured on Spectropolarimeter (Jasco, J-715). A total of 0.12-0.24 mg/mL of protein was incubated with 1-2 mM liposomes containing 4ME-PC/PE/PA for 40 mins prior to measurement. Liposomes/protein mixture was prepared in the buffer containing 20 mM Tris pH 7.5 and 150 mM NaCl. CD spectra were measured between 190 nm and 260 nm in increments of 1 nm, with a bandwidth of 50 nm and an averaging time of 1 min at 25 °C. 10 iterations of spectra were averaged and was reported into a mDeg, which was converted to molar ellipticity ($m° × M/(10 × L × C)$) of which unit is expressed in degree cm$^2$ dmol$^{-1}$. Molar ellipticity of each protein fragment in the absence or presence was plotted using GraphPad prism software.

### Artificial lipid droplet flotation assay
Artificial lipid droplets (ALDs) were prepared by mixing phospholipids and triacylglycerols (TAGs). Di-oleoyl phospholipids were dried under a nitrogen gas stream. Dried lipids were resuspended with lipid droplet flotation assay buffer containing 50 mM NaCl and 50 mM HEPES pH 8.0 and TAGs in the molar ratio of 2:5. ALDs were formed by repeating the cycles of vortex for 10 sec and rest for 10 sec. ALDs were further vortexed before use for the assay. One hundred microlitres of ALDs were mixed with 5 μL of human PLIN3 and 105 μL of lipid flotation assay buffer to give a final concentration of 5 μM protein and 1.5 mM phospholipids. The concentration of ALDs was determined by optical density at 600 nm. The mixture of human PLIN3 and ALDs was incubated for 1 h at 23 °C. To generate a sucrose gradient, 140 μL of a 75% sucrose solution was added to 210 μL of the protein-ALD mixture to give a final concentration of 30% sucrose. The resulting mixture (320 μL) was transferred to an ultra-centrifuge tube. Two hundred and sixty microlitres of 25% sucrose solution were applied on top of 30% sucrose/protein/ALDs mixture. Lastly, 60 μL of lipid flotation assay buffer was laid on the top. ALDs were floated by centrifugation at 76,000 x g for 3 h at 20 °C with a sucrose gradient of 30% (bottom), 25% (middle), and 0% (top). Three fractions of 100 μL from the top, 260 μL from the middle, and 280 μL from the bottom were collected and analyzed by SDS-PAGE.

### HDX-MS sample preparation
HDX reactions for PLIN3 deuterium pulse were conducted in a final reaction volume of 10 μL with a final concentration 2.12 μM PLIN3.

Exchange was carried out in triplicate for a single time point (3 sec at 0 °C). Hydrogen deuterium exchange was initiated by the addition of 9 μL of D2O buffer solution (20 mM HEPES pH 7.5, 100 mM NaCl) to 1 μL of protein, to give a final concentration of 84.9% D2O. Exchange was terminated by the addition of 60 μL acidic quench buffer at a final concentration 0.6 M guanidine-HCl and 0.9% formic acid. Samples were immediately frozen in liquid nitrogen at −80 °C. Fully deuterated samples were generated by first denaturing the protein in 3M guanidine for 1 h at 20 °C. Following denaturing, 9 μL of D2O buffer was added to the 1 μL of denatured protein and allowed to incubate for 10 mins at 20 °C before quenching with 0.6 M guanidine-HCl and 0.9% formic acid. Samples were immediately frozen in liquid nitrogen at −80 °C, and all timepoints were created and run in triplicate.

HDX reactions comparing PLIN3 in the presence of PA liposomes (60 mol% 4ME-PC, 20 mol% 4ME-PE, 20 mol% 4ME-PA) were conducted in a final reaction volume of 10 μL with a final protein concentration of 3 μM and final liposome concentration of 1 mM. Protein and liposomes were preincubated together for 2 mins at 20 °C before the addition of 7 μL D2O buffer solution (50 mM HEPES pH 8.0, 50 mM NaCl) for a final concentration of 63% D2O. Exchange was carried out for 3 sec, 30 sec, 300 sec, and 3000 sec, and exchange was terminated by the addition of 60 μL acidic quench buffer at a final concentration 0.6 M guanidine-HCl and 0.9% formic acid. Samples were immediately frozen in liquid nitrogen at −80 °C, and all time points were created and run in triplicate.

HDX reactions comparing PLIN3 in the presence of two different liposomes (60 mol% DOPC, 20 mol% DOPE, 20 mol% DAG liposomes and 80 mol% 4ME-PC, 20 mol% 4ME-PE liposomes) were conducted in a final reaction volume of 10 μL with a final protein concentration of 3 μM and final liposome concentration of 500 μM. Protein and liposomes were preincubated together for 2 mins at 20 °C before the addition of 8 μL D2O buffer solution (50 mM HEPES pH 8.0, 50 mM NaCl) for a final concentration of 72% D2O. Exchange was carried out for 3 sec, 30 sec, 300 sec, and 3000 sec, and exchange was terminated by the addition of 60 μL acidic quench buffer at a final concentration 0.6 M guanidine-HCl and 0.9% formic acid. Samples were immediately frozen in liquid nitrogen at −80 °C, and all timepoints were created and run in triplicate.

### HDX-MS Protein digestion and MS/MS data collection
Protein samples were rapidly thawed and injected onto an integrated fluidics system containing a HDx-3 PAL liquid handling robot and climate-controlled (2 °C) chromatography system (LEAP Technologies), a Dionex Ultimate 3000 UHPLC system, as well as an Impact HD QTOF Mass spectrometer (Bruker). The protein was run over either one (at 10 °C) or two (at 10 °C and 2 °C) immobilized pepsin columns (Trajan; ProDx protease column, 2.1 mm x 30 mm PDX.PP01-F32) at 200 μL/min for 3 mins. The resulting peptides were collected and desalted on a C18 trap column (Acquity UPLC BEH C18 1.7 mm column (2.1 x 5 mm); Waters 186003975). The trap was subsequently eluted in line with an ACQUITY 1.7 μm particle, $100 \times 1 \, mm^2$ C18 UPLC column (Waters), using a gradient of 3–35% B (Buffer A 0.1% formic acid; Buffer B 100% acetonitrile) over 11 mins immediately followed by a gradient of 35–80% over 5 mins. Mass spectrometry experiments were acquired over a mass range from 150 to 2200 m/z using an electrospray ionization source operated at a temperature of 200 °C and a spray voltage of 4.5 kV.

### HDX-MS peptide identification
Peptides were identified from the non-deuterated samples of PLIN3 using data-dependent acquisition following tandem MS/MS experiments (0.5 sec precursor scan from 150–2000 m/z; twelve 0.25 sec fragment scans from 150–2000 m/z). MS/MS datasets were analyzed using PEAKS7 (PEAKS), and peptide identification was carried out by using a false discovery based approach, with a threshold set to 1% using

a database of known contaminants found in Sf9 and Escherichia coli[85]. The search parameters were set with a precursor tolerance of 20 ppm, fragment mass error 0.02 Da, charge states from 1–8, leading to a selection criterion of peptides that had −10logP scores of 22.8 for the pulse experiment and 20.9 for the liposome experiment.

### HDX-MS mass analysis of peptide centroids and measurement of deuterium incorporation
HD-Examiner Software (Sierra Analytics) was used to automatically calculate the level of deuterium incorporation into each peptide. All peptides were manually inspected for correct charge state, correct retention time, appropriate selection of isotopic distribution, etc. Deuteration levels were calculated using the centroid of the experimental isotope clusters. Results are presented as relative levels of deuterium incorporation, with the only correction being applied correcting for the deuterium oxide percentage of the buffer utilized in the exchange (63 and 72%). For the experiment with a fully deuterated sample, corrections for back exchange were made by dividing the pulse %D value by the fully deuterated %D value and multiplying by 100. The raw %D incorporation for the fully deuterated sample is included in the source data, with the average back exchange being 33%, and ranging from 10–60%. Differences in exchange in a peptide were considered significant if they met all three of the following criteria: ≥5% change in exchange and ≥0.4 Da difference in exchange. The raw HDX data are shown in two different formats. Samples were only compared within a single experiment and were never compared to experiments completed at a different time with a different final D2O level. The data analysis statistics for all HDX-MS experiments are in Supplementary Table 1 according to the guidelines of[86]. The mass spectrometry proteomics data have been deposited to the ProteomeXchange Consortium via the PRIDE partner repository with the dataset identifier PXD025717[87].

### Dynamic light scattering
The size of liposomes was measured by dynamic light scattering (Brookhaven Instruments NanoBrook Omni), with or without full-length PLIN3. Liposomes were prepared in the buffer containing 50mM HEPES pH 8.0 and 50 mM NaCl by repeating the freeze/thaw cycles to prevent multilamellar vesicle formation. Each protein construct was incubated with liposomes in 1:200 of molar ratio at 25 °C for 40 mins. DLS was conducted using a scattering angle of 90°, and the liposome/protein mixture was equilibrated for 5 mins after loading into the instrument to get a uniform temperature and minimize any loading effects prior to measurement. The number average size distribution (%) was considered as a relative concentration of particles with a certain size. For analysis, measurements with large outlier peaks, which were suspected to be due to dust particles or aggregated vesicles, were discarded, and three runs that did not contain outlier peaks were used for the data analysis. The mode of the distribution (e.g., the size having the highest peak in the number average size distribution) were chosen from each of these three measurements and averaged to obtain the reported average diameter. A representative size distribution is shown for each sample.

### Site-directed spin-labeling
Recombinant PLIN3 cysteine-substituted proteins were purified using Ni-NTA resin in a buffer 500 mM NaCl, 20 mM Tris pH 7.5, 5% glycerol, and eluted in a buffer of 500 mM NaCl, 20 mM Tris pH 7.5, 5% Glycerol, 300 mM imidazole. Elution was spin-labeled with 1 μg/mL of MTSL (S-(1-oxyl-2,2,5,5-tetramethyl-2,5-dihydro-1H-pyrrol-3-yl) methyl methanesulfonothioate) (Santa Cruz Biotech) dissolved in acetonitrile at 4 °C overnight. The spin-labeled proteins were further purified by HiLoad 26/600 Superdex 200 pg column (GE life sciences) in a buffer of 150 mM NaCl, 20 mM Tris pH 7.5 to remove unreacted spin labels.

## Sample preparations for DEER spectroscopy

Protein samples in solutions were prepared at a total of 40–60 μM concentrations in 20 mM Tris at pH 7.5, 150 mM NaCl, and 40% glycerol. Liposome samples containing 10 μM protein and 3.33 mM lipid were prepared by mixing 30 μM protein stocks with 5 mM liposome solutions using 1 to 2 aliquots followed by 30 mins incubation at room temperature. The liposomes were freshly prepared by rehydration of lyophilized 80/20 mol% 4ME-PC/PE followed by seven freeze/thaw cycles. Freshly prepared 20 μL of spin-labeled samples were loaded into 1.8 mm inner diameter Pyrex sample tubes (Wilmad-LabGlass), frozen by plunging in liquid nitrogen, and stored in liquid nitrogen for PDS measurements.

## DEER data collection and analysis

PDS DEER measurements were performed at 60 K with a home-built Ku band pulse ESR spectrometer operating around 17.3 GHz[64,88]. A four-pulse DEER sequence[89] was employed using spin-echo detection π/2- and π-pulses with respective widths of 16 and 32 ns, the π-pulse for pumping was 16 ns. The detection frequency matched the peak at the low-field spectral edge, while pumping was performed at a lower by 70 MHz frequency, corresponding to the central maximum. A 32-step phase cycle[90] was applied to suppress unwanted contributions to the signal. Nuclear electron spin-echo envelope modulation (ESEEM) caused by surrounding protons was suppressed using the data from four measurements. In this method the initial pulse separations and the start of detection were advanced by 9.5 ns in subsequent measurements, i.e. by the quarter period of the 26.2 MHz nuclear Zeeman frequency for protons at 615 mT corresponding to the working frequency. The four signal traces were summed to achieve deep suppression of nuclear ESEEM[91].

The solution samples had phase memory times, $T_m$'s, of about 2.5 μs, so the data could be recorded up to 3 μs evolution time ($t_m$). Such evolution times did not provide sufficient decay of the dipolar coherence to the background level, providing however clear indication of the unstructured nature of the protein in solution. Any residual secondary structure could be probed by using multiple labeling sites, however, other techniques did not encourage this undertaking. For solutions, the DEER data were acquired in less than 12 h. Faster phase relaxation times (~1 μs) and low spin concentrations in liposome samples required signal averaging for ~24 h to facilitate good reconstruction by Tikhonov regularization using MATLAB[92,93].

For liposomes, time domain DEER data, $V(t)$, were recorded and preprocessed using standard approaches[62,65,89,94] before their reconstruction into distance distributions, $P(r)$'s. The first step is to remove signal decay caused by intermolecular dipole-dipole couplings, followed by subtracting the residual background from the spins whose partner was not flipped by the pump or missing. This was done, as usual, by fitting the latter points (about a half of the record) of $\ln[V(t)]$ to a low-order polynomial, usually a straight line for solutions (and often for liposomes), extrapolating it to zero time, and subtracting out from $\ln[V(t)]$; so that the antilog yields $u(t) = d(t) + 1$. Here $d(t)$ is the dipolar evolution representing a part. Once $u(t)$ is normalized as $u(t)/u(0)$, it serves as a typical form of DEER data presentation, while $v(t) = (u(t)-1)/u(0)$, gives background-free "dipolar" data to be converted to a distance distribution between spins in pairs. We used L-curve Tikhonov regularization[92] for distance reconstruction. The Tikhonov regularization utility allowed the selection of either the signal or its derivatives in the Tikhonov functional and the selection of the regularization parameter. The latter increases $P(r)$ smoothness at the expense of introducing some broadening.

Dipolar signal amplitude ("modulation depth") is given by $v(0)$. This would be accurate to the extent the background or $V(\infty)$, the asymptotic value of $V(t)$, is known. This depth, tabulated by calculations for typical pulse sequence setup and verified in numerous experiments, is a measure of a fraction of spins in pairs or

oligomers[69,95,96]. It is thus useful in the estimate of spin-labeling efficiency and protein concentration.

## Cell culture

Cos7 cells were maintained in Dulbecco's modified Eagle's medium (DMEM) supplemented with 10% heat-inactivated fetal bovine serum (Life Technologies), 4.5 g/L D-glucose, 0.1 g/L sodium pyruvate (Life Technologies) and 1% penicillin-streptomycin (Life Technologies). The cells were cultured at 37 °C under a 5% $CO_2$ atmosphere.

## Transfection

When indicated, Cos7 cells (60–70% confluence) plated into a 35 mm cell-culture Mattek dishes (with a glass coverslip at the bottom), (MatTek Corp. Ashland, MA). were transfected with 3 μg of plasmid DNA/ml using Polyethylenimine HCl MAX (Polysciences) following the manufacturer's instructions. For co-expressions, RFP-KDEL or GFP-PLIN3 constructs in equal concentrations (1.5–2 μg for each one) were transfected to cells 24 h prior observation.

## Giant unilamellar vesicles, GUVs

GUVs were prepared by electroformation following the protocol described in[97]. Phospholipids, dioleoyl phosphatidylcholine (PC) (70%) and dioleoyl phosphatidylethanolamine (PE) (29%), Rhodamine-DOPE (1% (w/w)), were purchased from Avanti Polar Lipids. The lipid mixture, in chloroform at 0.5 mM, was dried on an indium tin oxide (ITO)-coated glass plate. The lipid film was desiccated for 1 h. The chamber was sealed with another ITO- coated glass plate. The lipids were then rehydrated with a sucrose solution (275 mOsm). Electroformation was done using 100 Hz AC voltage at 1.0 to 1.4 Vpp and maintained for at least 1 h.

## Droplet-embedded vesicles (DEVs) preparation

Artificial LDs (aLDs) were prepared in HKM buffer: 50 mM HEPES, 120 mM Potassium Acetate, and 1 mM $MgCl_2$ (in Milli-Q water) at pH 7.4 and $275 \pm 15$ mOsm. To do so, 5 μL of the lipid oil solution (triolein or diacylglycerol purchased from Sigma) was added to 45 mL of HKM buffer and the mixture was sonicated as described in ref. 98. The resulting emulsion was then mixed with GUVs for five minutes under the rotator to generate DEVs[98]. DEVs were then placed on a glass coverslip pretreated with 10% (w/w) BSA and washed three times with buffer.

## Protein binding to DEVs

For the protein binding experiments, 50 μL HKM were deposited on the BSA-treated glass, 30 μL of the DEV solution added and 1.5 μL purified PLIN3 or PLIN3 fragments were added.

## Neutral lipid synthesis induction

Wherever relevant, cells were exposed for 1 h to 350 μM oleic acid (OA) coupled to bovine serum albumin (BSA) (0.2% w/v) to induce neutral lipids' synthesis. LipidTox DeepRed (Thermo Fischer), was used to visualize lipid droplets or membranes enriched in neutral lipids.

## Giant intra-cellular ER vesicle experiments

For GERV experiments, Cos7 cells were first transfected for 24 h with the indicated eGFP plasmids and RFP-KDEL. The culture media of the cells was next replaced by a hypotonic culture media (DMEM: $H_2O$, 1:20). The cells were then incubated at 37 °C, 5% $CO_2$ for 5 mins, to induce GERVs, and then they were visualized. Neutral lipids' synthesis was triggered by the addition of OA and Z-stacks imaging of different entire cells was performed before and 15 mins after OA administration, as described in ref. 77. For the DGAT1 (Sigma, PF-04620110) and DGAT2 (Sigma, PF-06424439) inhibitors used in Cos7, the dilution applied was 1/1000 for a final concentration of 3 μg/mL. In the GERV protocol, the inhibitors were added to the cell medium before cell transfection or during the hypotonic solution addition. The signal of PLIN3 or fragments thereof was quantified for tens of GERVs by depicting the mean signal on the

GERVs which was subtracted the bulk cytosolic signal; this resulted signal was normalized to the cytosolic signal.

## Structure prediction by AlphaFold and RoseTTAFold

Structure prediction of PLIN3 was carried out by AlphaFold and RoseTTAFold. Two different confidence values were generated (pLDDT from AlphaFold and rmsd from RoseTTAFold). These values were converted using PHENIX to get relative confidence level from the two softwares, AlphaFold and RoseTTAFold[99].

## Reporting summary

Further information on research design is available in the Nature Portfolio Reporting Summary linked to this article.

## Data availability

Mass spectrometry proteomics data are available in PRIDE with the dataset identifier PXD025717. Source data are provided with this paper.

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

## Acknowledgements
This work was supported by in part by the NIGMS (R35GM128666, MVA), the Sloan Research Foundation (MVA), the Natural Science and Engineering Research Council of Canada (Discovery Grant 2020-04241, JEB) with salary support from the Michael Smith Foundation for Health Research (Scholar award 17868, JEB), the ANR (18-CE11-0012-01-MOBIL, CE11-0032-02-LIPRODYN and 21-CE13-0014-LIPDROPER, ART), and the NSF (CBET 1903189 and DGE 1922639, SRB). ESR study, conducted at ACERT, was supported by NIH/NIGMS grants 1R24GM146107 and P41GM103521.

## Author contributions
Y.M.C. generated constructs for E. coli and mammalian cell expression, purified all proteins, conducted liposome sedimentation, artificial lipid droplet binding assays, C.D., D.L.S. with assistance from SF, prepared figures, and wrote the initial draft. D.A. conducted all DEV and cell culture experiments. KDF, MLJ, and BEM setup, analyzed, and/or prepared figures and text from the HDX-MS data. P.P.B. designed, acquired, analyzed, and prepared figures for all PDS-ESR data. J.E.B., A.R.T., J.H.F., S.R.B., and M.V.A. supervised the work and provided funding support.

## Competing interests
JEB reports personal fees from Scorpion Therapeutics and Olema Oncology; and research grants from Novartis. The remaining authors declare no competing interests.
