## [Peer Review File · Nature Communications]

Structural insights into perilipin 3 membrane association in response to diacylglycerol accumulation

Editorial Note: Parts of this Peer Review File have been redacted as indicated to remove third party material where no permission to publish were obtainedREVIEWER COMMENTS

Reviewer expertise:

Reviewer #1 (perilipin 3)

Reviewer #2 (lipid droplets)

Reviewer #3 (protein biophysics and bioinformatics, disordered proteins)

Reviewer #4 (DEER/ ESR / membrane proteins)

Reviewer #5 (HDX-MS)

Reviewer #1 (Remarks to the Author):

Strengths:

The deuterium exchange studies of isolated and membrane bound perilipin 3 are novel and provide new and exciting insight into the structure of this ubiquitously expressed lipid droplet binding protein. The deuterium exchange assay and following mass spec methods are clear and well described.

The finding that the N-terminus (PAT and 11-mer repeat region) binds preferentially to membranes containing DAG is novel and provides important information on the mode of lipid droplet binding by perilipin 3.

Overall the work appears to be carried out well is of high technical quality.

Weaknesses:

While the work is generally novel and appears to be well carried out I have significant questions and reservations on some aspects of this work. At the very least the authors need to discuss their data in more detail and depth (for example, what do the authors mean by membrane defects?) and the authors should also sight more relevant literature.

On page 5 the authors discuss the preferential binding of perilipin 3 to liposomes containing DAG and PA, and only to those species that have a dioleoyl hydrocarbon tail composition. The finding that

perilipin 3 binds preferentially to dioleoyl, versus palmitoyl-oleoyl acyl chains is NOT novel and has been published before in:

Titus, A. R., Ridgway, E. N., Douglas, R., Brenes, E. S., Mann, E. K., and Kooijman, E. E. (2021) The C-Terminus of Perilipin 3 Shows Distinct Lipid Binding at Phospholipid-Oil-Aqueous Interfaces, *Membranes*.

Additionally, others have shown that a related perilipin, perilipin 2, binds preferentially to lipids containing phosphatidylethanolamine, or in other words those that have negative spontaneous curvature. Published in:

Listenberger, L., Townsend, E., Rickertsen, C., Hains, A., Brown, E., Inwards, E. G., Stoeckman, A. K., Matis, M. P., Sampathkumar, R. S., Osna, N. A., and Kharbanda, K. K. (2018) Decreasing Phosphatidylcholine on the Surface of the Lipid Droplet Correlates with Altered Protein Binding and Steatosis, *Cells* 7

These works should be cited in the manuscript and can assist in the further discussion of the data. Especially in light of the fact that PA is also a membrane lipid with negative curvature, like DAG and PE. So why do the authors see preferential binding to PA and DAG, but not PE? What drives specificity? Can the authors further define membrane defects?

One aspect of the methods that needs further elaboration is the liposome binding assay. Figure 1 details these results and they are discussed in the methods, but no details on the liposomes themselves is given. Have the authors characterized their liposomes? I am speculating that the authors used MLVs (multilamellar liposomes) but they never explicitly mention this. Normally with liposome binding assays single bilayer liposomes such as LUVs or SUVs are used to gain deeper insight into lipid binding. These are always characterized using dynamic light scattering to assure a homogeneous population of liposomes. The MLV assay is inherently messy as much of the lipid is contained on the inside of the large multilamellar lamellar structures. Clear characterization of the liposomes used is needed!

What strikes me as odd is that the authors were able to contain 20 mol% of DAG in their liposomes without issues. It is well known that DAG is a fusogenic lipid that causes extensive fusion of membranes in liposomes. The authors would do well to review the relevant literature (and cite these). For example:

Reviews on DAG:

1) Felix M. Goni, and Alicia Alonso, (1999) Structure and functional properties of diacylglycerols in membranes, *Progress in Lipid Research*, 38, 1-48

2) Ilvia Carrasco, and Isabel Mérida (2007) Diacylglycerol, when simplicity becomes complex, Trends in Biochemical Sciences, 32, 27-36

DAG incorporated in liposomes causes extensive membrane fusion and isolation of DAG from the exterior membrane surface: Putta, P., Rankenbreg, J., Korver, R. A., van Wijk, R., Munnik, T., Testerink, C., and Kooijman, E. E. (2016) Phosphatidic acid binding proteins display differential binding as a function of membrane curvature stress and chemical properties, Biochim Biophys Acta 1858, 2709-2716.

On page 6 (line 145) the authors assert that the main function of perilipin 3 is to bind to emerging LDs. However, how well established is this and why don't the authors cite the relevant literature for this?

On page 9 (line 230) the authors talk about "severe membrane packing defects" but do not quantify, or discuss what they mean by this. Some measure of discussion is definitely in place here.

On this page also, the authors discuss that the helix bundle of perilipin 3 requires PA to bind to membranes but don't go into any further detail. It would be nice to have a general discussion as to what drives the helix bundle to bind PA as much is known about PA binding from the membrane literature.

The authors use AlphaFold and RosettaFold to look at the structure of the N-terminus of perilipin 3. However, there is no discussion on the significance of these results. When I run alpha fold the confidence level of the structure it produces for the N-terminus is extremely low, and that makes sense as there are few (if any) structures of similar sequences known. AlphaFold does not calculate a structure from basic principles, instead it uses an AI to come to a consensus structure. At least that is my understanding.

The reason I bring this up is that the authors show structures for sequences that are likely to be speculative at best. In fact, they have shown beautifully in their deuterium exchange experiments that the N-terminus is highly unstructured. They then go to show using the DEER experiments that the alphaFold structure is indeed incorrect. So why then show these data? I would be ok with these structures if they were fully discussed and confidence levels were shown and also properly addressed. Without this these data aren't very useful.

Additionally, (and this is relevant for the deuterium exchange data as well, where the following paper should be cited), it is well known that the structure of perilipin 3 in solution is highly extended (aka, unfolded):

Hynson, R. M., Jeffries, C. M., Trewthella, J., and Cocklin, S. (2012) Solution structure studies of monomeric human TIP47/perilipin-3 reveal a highly extended conformation, Proteins 80, 2046-2055.

In the methods the authors need to provide the optimized sequence for PLIN3. (see page 16 line 433)
Obviously this can (and should) go to the supplementary data file.

It would also be useful to mention the general yield of the protein expression and purification assays.
Does the incorporation of GFP lead to improved expression and purification or not?

Page 31, line 960, please provide accession number for PLIN3 (here or in the methods). This is the legend of Figure 2.

Page 33, line 1022, please provide accession numbers for PLIN 1, 2, 3, and 5.

Reviewer #2 (Remarks to the Author):

Structural data on PLINs is limited and data linking structure to function is rarer. This paper dissects 1) PLIN3's PAT and 11-mer domains lipid requirements for membrane binding. 2) lipid binding affects PLIN3 structure. This in vitro work is consistent with cell imaging studies indicating that PLIN3 binds DAG enriched membrane leaflets. These imaging studies, one of which was work from my lab, have weaknesses including not being quantitative, limited ability to manipulate or understand leaflet composition of the PLIN3 bound membranes and interference from other proteins (for example other proteins interaction with PLIN3). This study is not subject to these limitations. This study dissects the lipid compositions of membranes required for specific perilipin-3 domains to bind membranes and how lipid binding affects PLIN3 structure. This work makes meaningful contributions at the mechanistic and structural level. In addition, supplying a molecular mechanism for PLIN binding to DAG enriched membranes makes my previous and more notably the recent work of Khaddaj and Schneiter, more relevant. In short, this work contributes clear new findings and these findings both are relevant alone and puts other work into a more meaningful context. Specifically, the culmination of decades of work resulted in a model where PLIN1 is the adaptor that organizes the catabolism of fat. Specifically, PLIN1's metabolic state dependent phosphorylation controls PLIN1 structure and thereby binding of CGI-58. This binding is the core control of fat release from fat cells. Thus, PLIN1 can be considered an adaptor protein important in anabolism. To this point, other than parallels drawn between PLIN1 and other PLINs, little is known about the other PLINs. DAG being a precursor to the fat storage molecule TAG accumulates in membranes during fat storage. This work shows that PLIN3 binds DAG enriched membranes and this binding causes PLIN3 structure to change. Recent in vivo work suggests that PLIN3, in fact, organizes lipid droplet biogenesis (see link 1 below (Khadda and Schneiter, 2022)). Thus, this work in context suggests PLIN3 is an adaptor important in anabolism.

Suggestions:

-To increase accessibility to people not focused on the lipid aqueous interface, include in the paragraph introducing lipid binding motifs (lines 77-82) the idea that PLINs bind LD when the hydrophobic side of amphipathic helices gain access to the inside of membrane leaflets and these helices then are held inside by the hydrophobic ends the phospholipids. Then connect the ideas including words to the effect of “amphipathic helices imbed in the hydrophobic inside of the membrane exposed by disordered phospholipids”. The red shaded triangle was useful for me. I suggest calling attention to the triangle and this idea in the results.

-Fig 2. hydrogen deuterium exchange mass spectrometry. If I understand this method, stable and ordered structures do block the deuterium labeling and disordered and fluid structures allow access to the label and thus label well. Stating this more plainly would have been helpful to me and likely other readers.

-You show experimental data that indicate that the PAT domain extends into the repeats.

This region was defined ~ 2 decades ago with scant sequence data available. I doubt there was any data indicating it was a true domain. The author’s did well given the information they had. You have a lot more data. I suggest pointing out that 1-97 is not stable which suggests that it is not an independently folding domain. I think a stronger statement defining the PAT region as a domain and more precisely delineating it is warranted. Specifically, state that your data shows a functional domain between ~22-117. Cited exact range as indicated by your data.

I suggest leaving out the COS7 imaging experiments. In short, all the good things about the in vitro work I stated above are negated. Even though the images in this work are good. In fact, they are giant unilamellar vesicles striking and informative.

However, the COS7 data has at least 3 issues working against it:

I have found GFP labeled perilipins do not localize to lipid droplets as well as unlabeled perilipins.

In my study I found it difficult to get PLIN3 to bind to ER COS7. That is why we use OP9 cells.

ER is large, complicated and heterogenous. The RFP_KDEL is a luminal ER marker and thus preferentially labels the perinuclear rough ER. Whereas PLIN3 preferentially binds the smooth ER. It is not surprising that the localization with the ER is limited. See the supplemental material fig 1 in this work for details (<https://www.ncbi.nlm.nih.gov/pmc/articles/PMC3661117/>).

I suggest that you reference your experiments as data not shown and cite the papers in the links below. I only scanned this paper, but I think the first paper (likely published after this was submitted) shows a similar result than yours by imaging and I think compliments your work here. Given the lipid binding and the corresponding structure change in PLIN3, consider including in the discussion the idea that PLIN3 has an adaptor function in LD biogenesis.

<https://www.biorxiv.org/content/10.1101/2022.09.14.507979v1>

<https://pubmed.ncbi.nlm.nih.gov/19748893/>

<https://journals.biologists.com/jcs/article/126/22/5198/53872/Expression-of-oleosin-and-perilipins-in-yeast>

-Line 73 “PLIN3 is unique in that it is stable and not degraded in the absence of LDs.” I think this is also true for PLIN4 and PLIN5. Thus, I do not think the word “unique” is accurate.

-Line 657 “Wherever relevant, cells were exposed for 1 h to 350 μ M oleic acid (OA) coupled to bovine serum albumin (BSA) (1% vol/vol) to induce neutral lipids’ synthesis.” Pure BSA is not a liquid and the concentration of BSA is not given. Thus, I do not understand what “1% vol/vol” means. I describe albumin-bound oleate by giving the oleate concentration and the oleate to albumin ratio (example : Oleate was bound to BSA at a ratio of 5.5 oleate/albumin).

Reviewer #3 (Remarks to the Author):

This is an interesting and very important study with great potential. Conducted analyses are very impressive and the conclusions are rather well supported by the results of these analyses. However, in my view, some additional experiments are needed to further enhance this work.

1) There are numerous publicly available computational tools for evaluation of intrinsic disorder predisposition of query proteins based on their amino acid sequences. Conclusions on the intrinsically disordered nature of various regions in PLIN3 can be supported applying these tools and reporting resulting disorder profiles. The authors should also take into account that the accepted practice in the field is to use several disorder predictors to generate such disorder profiles. I am attaching a file that provides outputs of a couple of these tools.

2) The authors suggest that their protein undergoes folding induced by interaction with membranes and that the membrane binding induces a disorder/order transition of alpha helices

within the PAT domain and 11-mer repeats. In my view, they should use circular dichroism spectroscopy to show these conformational changes directly.

>sp|O60664|PLIN3_HUMAN Perilipin-3 OS=Homo sapiens OX=9606 GN=PLIN3 PE=1 SV=3
MSADGAEADGSTQVTVEEPVQQPSVVDVAVASMPVLSSTCDMVSAAYASTKESYPHIKTVCDAAEKGVRTLTAADVSG
AQPILSKLEPQIASASEYAHRLDKLEENLPILQQPTEKVLADTKELVSSKVSQAQEMVSSAKDTVATQLSEAVDAT
RGAVQSGVDKTKSVVTGGVQSVMGSRVSGVDTVLGKSEEWADNHLPLTDAELARIATSLDGFVDVASVQQQRQ
EQSYFVRLGSLSERLRQHAYEHSLGKLRATKQRAQEALLQLSQVLSLMETVKQGVDQKLVEGQEKLVHQMWLSWNQKQ
LQGPEKEPPKPEQVESRALTMFRDIAQQLQATCTSLGSSIQGLPTNVKDQVQQARRQVEDLQATFSSIHQDLSSS
ILAQSRERVASAREALDHMVEYVAQNTPVTVLWVGFAPGITEKAPEEK

PONDR: <http://www.pondr.com/>

D2P2: <https://d2p2.pro/search/sequence>

RIDAO: <https://ridao.app/>

[redacted]

Reviewer #4 (Remarks to the Author):

This manuscript provides detailed and systematic data on how lipid composition affects PLIN3 recruitment to membrane bilayers and lipid droplets and identifies potential structural changes that occur upon the membrane binding. The most important findings are that the TAG precursors phosphatidic acid and diacylglycerol (DAG) are responsible for recruitment of PLIN3 to membrane bilayers and a better-defined (vs. previous studies) expanded PAT domain that is responsible for a preferential binding to the DAG-enriched membranes. Overall, the experiments were carried out with care and would be of the interest to the Journal readership. Thus, I would like to recommend the publication after the following improvements are made:

(1) I think the manuscript title “Structure and dynamics of human perilipin membrane association” significantly overstates what was accomplished by the authors. Basically, what was shown is that there is a disorder-to-order transition of a part of a protein upon membrane binding and a modelling that is indicative of a triangular arrangement. However, I would not call these findings “structure ... of human perilipin membrane association” as the authors do not show any data on this protein-membrane complex. Secondly, dynamics is hardly discussed at all. The only thing shown was that H-D exchange is slow in presence of liposomes. But how do we know that PLIN3 is fully accessible and whether the H-D exchange and the quenching were not affected by multilamellar vesicles formed/present? The authors not even measured kinetics of the protein binding. To conclude, the authors are asked to revise the title accordingly.

(2) P.14: “The PAT domain is predicted to adopt a triangular tertiary structure by both AlphaFold and RoseTTAfold. Our PD-ESR distance measurements are consistent with these predictions but not identical.” – How was the structure prediction carried out? Did the authors consider effects of lipids/lipid bilayers? Where is the lipid bilayer in Fig. 4? The two predicted structures show very large differences for the interspin distances and also deviate significantly from the results of DEER experiment. I do not see how the reported “ PD-ESR distance measurements are consistent with these predictions”. The only conclusion that that could be made from these distance measurements for just 1 spin pair is that there is conformational change when lipids are present and that the change brings the labeled sites within the DEER distance range. The rest is just pure speculation. Note that the triangular structure is only based on modelling. It would take more than one distance measurement with more mutants to verify the triangular arrangement experimentally. Please rewrite this section accordingly.

(3) While the authors discuss the PLIN3 binding to liposomes in terms of membrane lipid packing and associated defects on a qualitative level, there is no discussion of effects of surface charge on binding. Thus, it remains unclear whether the surface charge plays any role in binding. Please discuss the results in terms what is known about surface charges of the membrane preparations employed in the experiments..

(4) Several minor issues:

- Please check that all the abbreviations are spelled out in the text.

- Lines 84-85: “Here, we examined ... and analyze...” – please choose either past or present.

- Lines 294-295: “They (measurements) do not show oscillations, which is frequently observed for membrane proteins. [62-65]”. It is not measurements but DEER traces that show oscillations. The latter are typically observed for spin pairs with narrow distance distributions, which are not necessarily related to membrane proteins. Please rewrite.

- Lines 306-307: “... recording field-swept echo and did not notice any broadening that could indicate a shorter range of distances (<15 Å)”. Note, that the field-swept echo could miss spectral broadening due short relaxation times of spins at short distances. Continuous wave spectra would be a better measure.

- Lines 280-281: If “PDS is a collection of several ...pulse ESR techniques”, why write “Our PD-ESR distance measurements” in Line 404? Perhaps, “DEER distance measurements”? There is no need for “Our”.

Reviewer #5 (Remarks to the Author):

Overall, the HDX-MS experiments, data, and reporting are of high quality. It is also good to see data have been uploaded to PRIDE. However, there are some issues with the folding/unfolding datasets and the membrane binding datasets having some differences in construction which partially undermines the author’s conclusions. The materials and methods section needs to be generally smartened up also. Comments below:

- 1) Would the authors comment as to why the 0.3s timepoint is foregone in the membrane binding events? Given that this would provide the best comparison for the membranes facilitating a disorder/order transition.
- 2) Related to the prior comment – the disorder/order transition experiments were conducted at pH 7, 100 mM NaCl, membrane binding at pH 8, 50 mM NaCl. Why? pH may also drive protein folding, and will of course impact solvent uptake rates, so it adds a layer of possible complexity in the interpretation of disorder/order transition and the membrane binding. Can the authors explain why these differences exist?
- 3) Lines 206-211. There seems to be quite a theoretical leap here with the interpretation of HDX-MS data and membrane residency. Firstly, the authors state two possible scenarios for an explanation of their observed HDX-MS data. However, a third scenario also exists – that the PAT domain/11-mer repeats section remains unfolded and inserts into the membrane i.e. folding and membrane binding may be separate events. Would the authors be able to conduct an additional experiment, such as circular dichromatism, on the 1-196 construct to see if membranes induce alpha-helicity?
- 4) Given the bimodal distributions, has there been an attempt to deconvolute the two populations to determine whether the relative proportions are in keeping with their model of membrane binding?
- 5) There is variation in the description of the pulse experiment as being conducted at either 10C (Line 167 & 959) or 00C (Line 498). Please correct as necessary.

- 6) Line 499. "20 mM HEPES pH 7" – other values are given to a higher degree of precision. I assume this is 7.0?
- 7) Line 500 – no volume of quench buffer is provided.
- 8) Line 499, 509, 519, 511, 520, 521, 533... μM not uM , the same for volumes. This is repeated throughout the materials and methods section. Additionally, the " μ " is occasionally italicised throughout the manuscript. I'm not trying to be a pedant here but it's all over the place.
- 9) Why were different liposome concentrations used between PA and the DAG conditions used?
- 10) Line 521 – why was a different pH used in the DOPC liposomes (pH 7.0) cf. pH 8 for PA liposomes used? Is this a typo? It will surely affect the overall exchange kinetics?
- 11) 539 - 200 °C
- 12) No description of how the back-correction for the fully deuterated dataset is provided.

We thank the 5 reviewers for their constructive comments and provide a point-by-point response below. Accompanying our revised submission is a word document with track changes as a reference.

Reviewer #1 (Remarks to the Author):

Strengths:

The deuterium exchange studies of isolated and membrane bound perilipin 3 are novel and provide new and exciting insight into the structure of this ubiquitously expressed lipid droplet binding protein. The deuterium exchange assay and following mass spec methods are clear and well described.

The finding that the N-terminus (PAT and 11-mer repeat region) binds preferentially to membranes containing DAG is novel and provides important information on the mode of lipid droplet binding by perilipin 3.

Overall the work appears to be carried out well is of high technical quality.

Thank you for noting several strengths and novel findings in our manuscript.

Weaknesses:

While the work is generally novel and appears to be well carried out I have significant questions and reservations on some aspects of this work. At the very least the authors need to discuss their data in more detail and depth (for example, what do the authors mean by membrane defects?) and the authors should also sight more relevant literature.

Thank you for raising these important points. We have now added a general explanation about membrane packing defects in the introduction.

Specifically, we add the following text in lines #83-88: *“The binding of amphipathic helices, such as those found in the 11-mer repeats, to membrane interfaces is greatly influenced by the level or presence of phospholipid packing defects [34, 35]. The degree of packing is determined by the type of lipid present (as indicated in the triangle in Fig. 1A). When it comes to the oil-water interface of LDs, the recruitment of amphipathic helices is greater when the level of phospholipid packing is lower [35, 36], i.e. when there are more packing defects.”*

On page 5 the authors discuss the preferential binding of perilipin 3 to liposomes containing DAG and PA, and only to those species that have a dioleoyl hydrocarbon tail composition. The finding that perilipin 3 binds preferentially to dioleoyl, versus palmitoyl-oleoyl acyl chains is NOT novel and has been published before in:

Titus, A. R., Ridgway, E. N., Douglas, R., Brenes, E. S., Mann, E. K., and Kooijman, E.

E. (2021) The C-Terminus of Perilipin 3 Shows Distinct Lipid Binding at Phospholipid-Oil-Aqueous Interfaces, Membranes.

Thank you for pointing out this relevant paper. In our data, we found full length PLIN3 did not bind any PO liposomes, and can bind DO liposomes but only in the presence of DAG or PA. This indicates the membrane environment generated by DOPC and DOPE is not sufficient, but is required to recruit PLIN3 to liposomes containing DAG or PA. While the experiments described in Titus et al were conducted on a monolayer of phospholipids, their findings are still in line with the binding preference of PLIN3 against DO containing liposomes (i.e. bilayers), which we observed in our study (note that monolayers and bilayers do not have the same properties, Biophys J, 2021. 120(24): p. 5491-5503). We now cite the paper indicated by the referee and have adjusted the text as appropriate.

Specifically we edited the following sentence in lines #119-124: *“Initial liposome co-sedimentation experiments varied the ratio of neutral phospholipids phosphatidylcholine (PC) and phosphatidylethanolamine (PE), as PC and PE represent the major lipids on both the cytoplasmic face of the ER and surface of LDs, PE increases both PLIN2 binding to liposomes [43] and PLIN3 insertion into mixed lipid monolayers at phospholipid-oil interfaces [44] and the ratio of PC to PE has previously been shown to regulate protein distribution on LDs [43, 45].”*

Additionally, others have shown that a related perilipin, perilipin 2, binds preferentially to lipids containing phosphatidylethanolamine, or in other words those that have negative spontaneous curvature. Published in:

Listenberger, L., Townsend, E., Rickertsen, C., Hains, A., Brown, E., Inwards, E. G., Stoeckman, A. K., Matis, M. P., Sampathkumar, R. S., Osna, N. A., and Kharbanda, K. K. (2018) Decreasing Phosphatidylcholine on the Surface of the Lipid Droplet Correlates with Altered Protein Binding and Steatosis, Cells 7

Thank you for pointing out the experiment in Listenberger et al, which demonstrated increased binding of PLIN2 to liposomes with increased concentration of PE (specifically a ratio of PC:PE of 1.5:1; 40mol% PE). We now mention this finding in the text in lines #121-124: *“... PE increases both PLIN2 binding to liposomes [43] and PLIN3 insertion into mixed lipid monolayers at phospholipid-oil interfaces [44] and the ratio of PC to PE has previously been shown to regulate protein distribution on LDs [43, 45].”* Unlike PLIN2, we observed that increased concentrations of PE, were not sufficient to induce PLIN3 binding, even up to 60mol% PE. This is most likely due to differences in membrane binding between PLIN2 and PLIN3.

These works should be cited in the manuscript and can assist in the further discussion of the data. Especially in light of the fact that PA is also a membrane lipid with negative curvature, like DAG and PE. So why do the authors see preferential binding to PA and

DAG, but not PE? What drives specificity? Can the authors further define membrane defects?

For the questions: why do we see PA and DAG binding, but not PE? What drives specificity, and defining membrane packing defects? These are all interesting points with major implications for PLIN3 and the function of other perilipins. We have updated our discussion to try to address these points. Some important points we considered: 1) DAG is much more negatively curved than PA and PE. However, PE is less negatively curved than PA. Thus, from a curvature standpoint: DAG>>PE>PA, and 2) PE increases the charge of PA on membrane.

[redacted]

Table from Zanghellini et al. 2011 depicting the apparent area occupied by different lipids. The larger this area, the wider the packing defect.

40% PE would introduce more defects than 20%PA and yet binding was more significant with PA. Therefore, PLIN3 membrane association might not be solely about lipid packing defects, but physical selective interactions between PLIN3 and DAG/PA.

If it were solely about packing defects then PA would have introduced larger packing defects than PE (PA apparent area bigger than PE) and, therefore, yield higher protein recruitment. The only possibility that could explain such a situation would be via PA-PA head repulsions by charges (as proposed with PS), which could happen if too much PA is present in a membrane. We disfavor this hypothesis because PA is a non-membrane lipid and can be present at a maximum of 20% (ben Mbarek et al. 2017) (PA was used at 20% and PE at 40%) and is therefore diluted by other phospholipids. Hence, PA-PA repulsions would be irrelevant.

New text in the discussion was added as follows in lines #441-454:

“PA is another TAG precursor that is present at sites where LD originate [17] and can bind to seipin [16]. In addition to DAG, it was found to have a significant impact on the recruitment of PLIN3 to membranes in vitro. Thus, it may increase the translocation of PLIN3 to early LD formation sites. These two TAG precursors appear to provide specificity for the association of PLIN3 with membranes. From a curvature standpoint, DAG has a more negative curvature compared to PA and PE, whereas PE has a more negative curvature than PA [80]. This suggests that membrane curvature cannot solely account for the major role of PA in PLIN3 membrane binding specificity. However, from a surface charge perspective, PA and PE together may act synergistically by increasing charge of PA on the membrane [46, 81, 82]. In this scenario, PLIN3 recruitment to LD nucleation sites could be enhanced by specific recognition of PA, potentially through the 4-helix bundle. Therefore, PLIN3 membrane association may not only be determined by membrane packing defects, but could also involve selective physical interactions between

PLIN3 and DAG or PA. This idea is also supported by a previous study that found that the LD binding properties of PLINs are sensitive to the polar residue composition of their amphipathic helices [83].”

One aspect of the methods that needs further elaboration is the liposome binding assay. Figure 1 details these results and they are discussed in the methods, but no details on the liposomes themselves is given. Have the authors characterized their liposomes? I am speculating that the authors used MLVs (multilamellar liposomes) but they never explicitly mention this. Normally with liposome binding assays single bilayer liposomes such as LUVs or SUVs are used to gain deeper insight into lipid binding. These are always characterized using dynamic light scattering to assure a homogeneous population of liposomes. The MLV assay is inherently messy as much of the lipid is contained on the inside of the large multilamellar lamellar structures. Clear characterization of the liposomes used is needed!

We generated liposomes using repeated freeze/thaw cycles and now report data characterizing these liposomes using dynamic light scattering. This data is found in a new supplemental figure 1 and we have updated the main text and results accordingly. We cannot exclude the presence of multilamellar liposomes but, based on the DLS characterization showing overall integrated liposome area and given that the same amount of lipids was used on each case, we believe that all liposomes were similar in their physical characteristics. In the presence of DAG, we did observe larger diameter vesicles, which is most likely due to the fusogenic property of DAG as mentioned below by the reviewer. The addition of PA abrogates but does not eliminate fusion, and the addition of PLIN3 shifts the size distribution to smaller diameter vesicles, consistent with an ability of PLIN3 to remodel membranes. While use of SUVs would have resulted in cleaner experiments, our results showing recruitment by DAG are consistent with our DEV and cellular experiments, as well supported by other independent studies.

What strikes me as odd is that the authors were able to contain 20 mol% of DAG in their liposomes without issues. It is well known that DAG is a fusogenic lipid that causes extensive fusion of membranes in liposomes. The authors would do well to review the relevant literature (and cite these). For example:

Reviews on DAG:

1) Felix M. Goni, and Alicia Alonso, (1999) Structure and functional properties of diacylglycerols in membranes, *Progress in Lipid Research*, 38, 1-48

2) silvia Carrasco, and Isabel Mérida (2007) Diacylglycerol, when simplicity becomes complex, *Trends in Biochemical Sciences*, 32, 27-36

DAG incorporated in liposomes causes extensive membrane fusion and isolation of DAG from the exterior membrane surface: Putta, P., Rankenberg, J., Korver, R. A., van Wijk, R., Munnik, T., Testerink, C., and Kooijman, E. E. (2016)

Phosphatidic acid binding proteins display differential binding as a function of membrane curvature stress and chemical properties, *Biochim Biophys Acta* 1858, 2709-2716.

We now cite the Putta et al paper, which noted ~25% of DAG containing vesicles containing a lamellar like structure by EM. As noted above, we did observe larger diameter vesicles with 20 mol% DAG, which was likely due to similar vesicle fusion as noted by Putta et al. This property was reduced by the presence of PA and the addition of PLIN3 shifted the population distribution of vesicles to much smaller values.

On page 6 (line 145) the authors assert that the main function of perilipin 3 is to bind to emerging LDs. However, how well established is this and why don't the authors cite the relevant literature for this?

The function of PLIN3 binding to emerging LDs is extremely well characterized and it is conserved among species. To clarify this point, we have edited the introduction as follows in lines #76-78: *"This allows PLIN3 to translocate from the cytoplasm to sites of early LD formation, where PLIN3 is well established to act as a marker for the biogenesis of early LDs across species [10, 28, 29]."*

On page 9 (line 230) the authors talk about "severe membrane packing defects" but do not quantify, or discuss what they mean by this. Some measure of discussion is definitely in place here.

We have attempted to introduce membrane packing defects in more detail in the text above this sentence.

We now elaborate with updated text now reading in lines #248-252: *"We concluded that the 4-helix bundle of PLIN3 can unfold and bind membranes, but only under very specific condition such as the presence of accumulated PA on the membrane having packing defects induced by 4ME-PC/PE. This conclusion is supported by the PA accumulation at the nascent LD formation site where lipid packing defects occur [17]."*

On this page also, the authors discuss that the helix bundle of perilipin 3 requires PA to bind to membranes but don't go into any further detail. It would be nice to have a general discussion as to what drives the helix bundle to bind PA as much is known about PA binding from the membrane literature.

While we agree the specific PA binding by the bundle is an interesting point, we prefer to keep our discussion of these findings succinct at the moment. We have added the following sentence to the discussion, where we do speculate about the relevance of PA binding by the 4-helix bundle region of PLIN3.

New text now reads in lines #448-454: *“However, from a surface charge perspective, PA and PE together may act synergistically by increasing charge of PA on the membrane [46, 81, 82]. In this scenario, PLIN3 recruitment to LD nucleation sites could be enhanced by specific recognition of PA, potentially through the 4-helix bundle. Therefore, PLIN3 membrane association may not only be determined by membrane packing defects, but could also involve selective physical interactions between PLIN3 and DAG or PA. This idea is also supported by a previous study that found that the LD binding properties of PLINs are sensitive to the polar residue composition of their amphipathic helices [83].”*

The authors use AlphaFold and RosettaFold to look at the structure of the N-terminus of perilipin 3. However, there is no discussion on the significance of these results. When I run alpha fold the confidence level of the structure it produces for the N-terminus is extremely low, and that makes sense as there are few (if any) structures of similar sequences known. Alphafold does not calculate a structure from basic principles, instead it uses an AI to come to a consensus structure. At least that is my understanding.

The reason I bring this up is that the authors show structures for sequences that are likely to be speculative at best. In fact, they have shown beautifully in their deuterium exchange experiments that the N-terminus is highly unstructured. They then go to show using the DEER experiments that the alphafold structure is indeed incorrect. So why then show these data? I would be ok with these structures if they were fully discussed and confidence levels we shown and also properly addressed. Without this these data aren't very useful.

Thank you for bringing this up. We apologize for not showing the confidence level in the previous predicted structures (please see updated Fig. 4). This is clearly important, and we now show this in an updated Figure 4A. In addition, we also provide a predicted aligned error (PAE) plot to show inter-domain accuracy of AlphaFold prediction. We have also edited the discussion of the DEER experiment results, with an emphasis that the predicted DEER distance measurements are not consistent with an extended alpha helix, and are more in line with the triangular arrangement of helices predicted by both AlphaFold and RoseTTAFold.

The next text now read as follows in lines #466-474: *“The PAT domain is predicted to adopt a triangular tertiary structure by both AlphaFold and RoseTTAFold. The DEER distance measurements and CW ESR data are not identical to these predictions, but do indicate that when bound to membranes the PAT domain adopts a folded domain. This conclusion is supported by our HDX-MS results that found the peptides within the PAT domain display longer protection times from H-D exchange, which could be due to either a tertiary structure more resistant to unfolding or a longer membrane residency time. We note that the membrane bound PAT domain structure is likely dynamic and additional distance measurements at distinct sites are needed to verify the accuracy of the predicted triangular structures.”*

Additionally, (and this is relevant for the deuterium exchange data as well, where the following paper should be cited), it is well known that the structure of perilipin 3 in solution is highly extended (aka, unfolded):

Hynson, R. M., Jeffries, C. M., Trehella, J., and Cocklin, S. (2012) Solution structure studies of monomeric human TIP47/perilipin-3 reveal a highly extended conformation, *Proteins* 80, 2046-2055.

Thank you. We now cite this paper in the HDX-MS section; mentioning prior structural studies of PLIN3.

In the methods the authors need to provide the optimized sequence for PLIN3. (see page 16 line 433) Obviously this can (and should) go to the supplementary data file. It would also be useful to mention the general yield of the protein expression and purification assays. Does the incorporation of GFP lead to improved expression and purification or not?

We now provide the nucleotide sequence of the codon-optimized PLIN3 construct (Supplemental info) and note that altering the DNA sequence did not introduce any changes to the amino acid sequence. Yields are now mentioned in the methods section.

Page 31, line 960, please provide accession number for PLIN3 (here or in the methods). This is the legend of Figure 2.

UniProt ID: O60664. This is now added to the legend in Figure. 2B.

Page 33, line 1022, please provide accession numbers for PLIN 1, 2, 3, and 5.

UniProt ID: (O60240, Q99541, O60664, Q00G26). These are now added to the legend in Figure. 3A.

Reviewer #2 (Remarks to the Author):

Structural data on PLINs is limited and data linking structure to function is rarer. This paper dissects 1) PLIN3's PAT and 11-mer domains lipid requirements for membrane binding. 2) lipid binding affects PLIN3 structure. This in vitro work is consistent with cell imaging studies indicating that PLIN3 binds DAG enriched membrane leaflets. These imaging studies, one of which was work from my lab, have weaknesses including not being quantitative, limited ability to manipulate or understand leaflet composition of the PLIN3 bound membranes and interference from other proteins (for example other proteins interaction with PLIN3). This study is not subject to these limitations. This study dissects the lipid compositions of membranes required for specific perilipin-3 domains to bind membranes and how lipid binding affects PLIN3 structure. This work makes meaningful contributions at the mechanistic and structural level. In addition, supplying a

molecular mechanism for PLIN binding to DAG enriched membranes makes my previous and more notably the recent work of Khaddaj and Schneider, more relevant. In short, this work contributes clear new findings and these findings both are relevant alone and puts other work into a more meaningful context. Specifically, the culmination of decades of work resulted in a model where PLIN1 is the adaptor that organizes the catabolism of fat. Specifically, PLIN1's metabolic state dependent phosphorylation controls PLIN1 structure and thereby binding of CGI-58. This binding is the core control of fat release from fat cells. Thus, PLIN1 can be considered an adaptor protein important in anabolism. To this point, other than parallels drawn between PLIN1 and other PLINs, little is known about the other PLINs. DAG being a precursor to the fat storage molecule TAG accumulates in membranes during fat storage. This work shows that PLIN3 binds DAG enriched membranes and this binding causes PLIN3 structure to change. Recent in vivo work suggests that PLIN3, in fact, organizes lipid droplet biogenesis (see link 1 below (Khadda and Schneider, 2022)). Thus, this work in context suggests PLIN3 is an adaptor important in anabolism.

Thank you for noting the importance of our manuscript and how it incorporates and puts into context all the prior work on PLIN membrane binding and function.

Suggestions:

-To increase accessibility to people not focused on the lipid aqueous interface, include in the paragraph introducing lipid binding motifs (lines 77-82) the idea that PLINs bind LD when the hydrophobic side of amphipathic helices gain access to the inside of membrane leaflets and these helices then are held inside by the hydrophobic ends the phospholipids. Then connect the ideas including words to the effect of "amphipathic helices imbed in the hydrophobic inside of the membrane exposed by disordered phospholipids". The red shaded triangle was useful for me. I suggest calling attention to the triangle and this idea in the results.

Thank you for the suggestion. Now we introduce the interaction between amphipathic helix and exposed hydrophobic acyl chain of membrane by relating the potential packing defects of different phospholipid species. We specifically add new sentences as follows in lines #83-88: *"The binding of amphipathic helices, such as those found in the 11-mer repeats, to membrane interfaces is greatly influenced by the level or presence of phospholipid packing defects [34, 35]. The degree of packing is determined by the type of lipid present (as indicated in the triangle in Fig. 1A). When it comes to the oil-water interface of LDs, the recruitment of amphipathic helices is greater when the level of phospholipid packing is lower [35, 36], i.e. when there are more packing defects."*

-Fig 2. hydrogen deuterium exchange mass spectrometry. If I understand this method, stable and ordered structures do block the deuterium labeling and disordered and fluid structures allow access to the label and thus label well. Stating this more plainly would

have been helpful to me and likely other readers.

Thank you for helping us make our manuscript more accessible to a broad audience. We have updated the introduction of HDX-MS to now read as follows in lines #178-184: *“For these experiments, we used hydrogen deuterium exchange mass spectrometry (HDX-MS), which measures the exchange of amide hydrogens with deuterated solvent. This method acts as a readout for protein conformational dynamics with regions that form secondary structures undergoing slower deuterium exchange than disordered regions, which lack intramolecular hydrogen bonds and secondary structure [50, 51]. A brief pulse of deuterated solvent is useful for identifying regions within a protein that lack structure compared to ordered regions [52, 53].”*

-You show experimental data that indicate that the PAT domain extends into the repeats.

This region was defined ~ 2 decades ago with scant sequence data available. I doubt there was any data indicating it was a true domain. The author's did well given the information they had. You have a lot more data. I suggest pointing out that 1-97 is not stable which suggests that it is not an independently folding domain. I think a stronger statement defining the PAT region as a domain and more precisely delineating it is warranted. Specifically, state that your data shows a functional domain between ~22-117. Cited exact range as indicated by your data.

Thank you. While we were originally hesitant to make too strong of a statement, we have now updated our manuscript to suggest a more definitive conclusion based on our results. The text in the results now reads in lines #293-296: *“Taken together, the data suggests that the PAT region does form a domain that spans residues 22-116 in PLIN3. Notably, this expanded PAT domain is capable of membrane binding and displays a preference for binding DAG enriched membranes, while the 11-mer repeats of PLIN3 do not.”*

I suggest leaving out the COS7 imaging experiments. In short, all the good things about the in vitro work I stated above are negated. Even though the images in this work are good. In fact, they are giant unilamellar vesicles striking and informative.

While we appreciate the strong support of our in vitro experiments and acknowledge cellular experiments contain other factors that must be considered, we would like to include the COS7 data, as we believe it provides some important context to the in vitro work. We provide below an individual response to each point raised by the reviewer.

However, the COS7 data has at least 3 issues working against it:

I have found GFP labeled perilipins do not localize to lipid droplets as well as unlabeled perilipins.

This finding by the reviewer is noted. However, in the figures from our manuscript below, we identified a clear GFP-PLIN3 signal on lipid droplets in Cos7 cells (upper panels). It is only when we blocked DGATs (promoting DAG accumulation) that we found Plin3 on the Giant ER Vesicles (lower right panel).

In my study I found it difficult to get PLIN3 to bind to ER COS7. That is why we use OP9 cells.

Under intact cells, only PLIN1 can be easily seen in the ER because it is an ER membrane protein (please see Ajjaji 2019: DOI: 10.1091/mbc.E18-08-0534 and Ajjaji, 2022: DOI: 10.1111/tra.12825). PLIN2 and 3 do not localize to the ER under normal conditions in intact cells. When lipogenesis is triggered, PLIN3 (possibly PLIN2 also) rapidly localize to the ER, at regions enriched in DAG. However, because DAG is rapidly converted to TAG, it is difficult to see PLIN3. The rate of conversion of DAG will therefore determine how easily one can see PLIN3 in the ER during lipogenesis. This will depend on the expression and activity of DGATs, and other factors, which might differ between Cos-7 and OP9 cells.

A way to smooth out the discrepancy between cell lines and favor PLIN3 ER binding is by accumulating DAG in the ER, which we did by blocking DGATs. Please note: under normal lipogenic conditions, PLIN3 did not bind to the Giant ER Vesicles; but when DGATs were blocked, it bound.

ER is large, complicated and heterogenous. The RFP_KDEL is a luminal ER marker and thus preferentially labels the perinuclear rough ER. Whereas PLIN3 preferentially binds the smooth ER. It is not surprising that the localization with the ER is limited. See the supplemental material fig 1 in this work for details (<https://www.ncbi.nlm.nih.gov/pmc/articles/PMC3661117/>).

We thank the referee for pointing this out. However, we are not working on intact cells. In our study, the ER is swollen, which leads to the formation of micrometric ER vesicles, containing all ER lipids and proteins, and is suitable for imaging. We successfully showed that we can induce PLIN3 recruitment to these vesicles by blocking DGATs, which should promote DAG accumulation. Such an observation is consistent with our in vitro findings.

I suggest that you reference your experiments as data not shown and cite the papers in the links below. I only scanned this paper, but I think the first paper (likely published after this was submitted) shows a similar result than yours by imaging and I think compliments your work here. Given the lipid binding and the corresponding structure change in PLIN3, consider including in the discussion the idea that PLIN3 has an adaptor function in LD biogenesis.

<https://www.biorxiv.org/content/10.1101/2022.09.14.507979v1>

<https://pubmed.ncbi.nlm.nih.gov/19748893/>

<https://journals.biologists.com/jcs/article/126/22/5198/53872/Expression-of-oleosin-and-perilipins-in-yeast>

Thank you. We have incorporated the results from these manuscripts into an expanded discussion about the potential active role of PLIN3 in LD biogenesis.

-Line 73 “PLIN3 is unique in that it is stable and not degraded in the absence of LDs.” I think this is also true for PLIN4 and PLIN5. Thus, I do not think the word “unique” is accurate.

We have deleted the word “unique”.

-Line 657 “Wherever relevant, cells were exposed for 1 h to 350 μ M oleic acid (OA) coupled to bovine serum albumin (BSA) (1% vol/vol) to induce neutral lipids’ synthesis.” Pure BSA is not a liquid and the concentration of BSA is not given. Thus, I do not understand what “1% vol/vol” means. I describe albumin-bound oleate by giving the oleate concentration and the oleate to albumin ratio (example : Oleate was bound to BSA at a ratio of 5.5 oleate/albumin).

We apologize for the typo. It now reads: “Wherever relevant, cells were exposed for 1h to 350 μ M oleic acid (OA) coupled to bovine serum albumin (BSA) (0.2% weight/volume).”

Reviewer #3 (Remarks to the Author):

This is an interesting and very important study with great potential. Conducted analyses are very impressive and the conclusions are rather well supported by the results of these analyses. However, in my view, some additional experiments are needed to further enhance this work.

1) There are numerous publicly available computational tools for evaluation of intrinsic disorder predisposition of query proteins based on their amino acid sequences. Conclusions on the intrinsically disordered nature of various regions in PLIN3 can be supported applying these tools and reporting resulting disorder profiles. The authors should also take into account that the accepted practice in the field is to use several disorder predictors to generate such disorder profiles. I am attaching a file that provides outputs of a couple of these tools.

Thank you for providing these tools. We have already provided experimental evidence using HDX-MS but will keep in mind these tools for future studies.

2) The authors suggest that their protein undergoes folding induced by interaction with membranes and that the membrane binding induces a disorder/order transition of alpha helices within the PAT domain and 11-mer repeats. In my view, they should use circular dichroism spectroscopy to show these conformational changes directly.

Thank you for the excellent suggestion. We have carried out CD experiments of PLIN3 alone and in the presence of liposomes. This new data is included in a new main text figure and described in the main text.

The new paragraph reads as follows in lines #299-312: “*We first sought to confirm the effects of liposome binding on secondary structure of PLIN3 by circular dichroism (CD) (Fig. S5). Liposomes were prepared with 4ME-PC/PE/PA, which recruited all the PLIN3 constructs (full length, PAT, 11-mer repeats, PAT/11-mer repeats, 4-helix bundle) in the previous liposome co-sedimentation experiments. For full length PLIN3, the presence of liposomes increased overall helicity as observed by an increased negative peak around 222nm in the CD spectra. In comparison, the CD spectra of the 4-helix bundle was largely unaffected by the presence of liposomes and was consistent with a stable alpha helical structure. In contrast, the PAT/11-mer repeats adopted a mostly random coil structure in solution with a negative peak around 200nm, and a shift to alpha helices in the presence of liposomes as indicated by a large negative peak at 222nm. Liposomes induced similar changes in the CD spectra for both the PAT domain and 11-mer repeats alone. Taken together, this confirms that the increase in helicity observed in full length*

PLIN3 by membranes was due to the PAT/11-mer repeats undergoing a disorder/alpha helical transition. This is in line with our HDX-MS results and prior studies of PLIN2 and PLIN3 fragments [55, 59].”

Reviewer #4 (Remarks to the Author):

This manuscript provides detailed and systematic data on how lipid composition affects PLIN3 recruitment to membrane bilayers and lipid droplets and identifies potential structural changes that occur upon the membrane binding. The most important findings are that the TAG precursors phosphatidic acid and diacylglycerol (DAG) are responsible for recruitment of PLIN3 to membrane bilayers and a better-defined (vs. previous studies) expanded PAT domain that is responsible for a preferential binding to the DAG-enriched membranes. Overall, the experiments were carried out with care and would be of the interest to the Journal readership. Thus, I would like to recommend the publication after the following improvements are made:

(1) I think the manuscript title “Structure and dynamics of human perilipin membrane association” significantly overstates what was accomplished by the authors. Basically, what was shown is that there is a disorder-to-order transition of a part of a protein upon membrane binding and a modelling that is indicative of a triangular arrangement. However, I would not call these findings “structure ... of human perilipin membrane association” as the authors do not show any data on this protein-membrane complex. Secondly, dynamics is hardly discussed at all. The only thing shown was that H-D exchange is slow in presence of liposomes. But how do we know that PLIN3 is fully accessible and whether the H-D exchange and the quenching were not affected by multilamellar vesicles formed/present? The authors not even measured kinetics of the protein binding. To conclude, the authors are asked to revise the title accordingly.

Thank you for the constructive comments. We agree that more distance constraints are needed to arrive at a better view on the structure of PLIN3 to membranes. This will be considered strongly for future studies. While only two double mutants were measured in liposomes, these do provide quantitative distance restraints, which is relevant to PLIN3 structure. These initial steps are in the right direction to fully define PLIN3 structure, but we agree that a title change is warranted. Our new title is: “Structural insights into perilipin 3 membrane association in response to diacylglycerol accumulation”

(2) P.14: “The PAT domain is predicted to adopt a triangular tertiary structure by both AlphaFold and RoseTTAFold. Our PD-ESR distance measurements are consistent with these predictions but not identical.” – How was the structure prediction carried out? Did the authors consider effects of lipids/lipid bilayers? Where is the lipid bilayer in Fig. 4?

For structure predictions, we used the RoseTTAFold webserver or downloaded the publicly available AlphaFold structure (UniProt: O60664) from the AlphaFold Protein

Structure Database. We now include descriptions for structure prediction in the methods section. The effects of lipids/lipid bilayer were not considered. We will consider MD simulations with these parameters in future studies but consider this outside the scope of this study.

The hydrophobic surface of predicted PAT domain is now shown in the Figure 4B. In addition, we also provide a predicted aligned error (PAE) plot to show inter-domain accuracy of AlphaFold prediction (Figure 4C). We now note in the figure legend (Figure. 4) as follows: **B)** *The hydrophobic surface of predicted PAT domain was depicted. The hydrophobic face of the helices is facing towards the reader.*

C) *The PAE (Predicted Aligned Error) value of AlphaFold2 predicted full length PLIN3 was plotted by ChimeraX and shown as an interactive 2D plot (bottom, right panel)."*

The two predicted structures show very large differences for the interspin distances and also deviate significantly from the results of DEER experiment. I do not see how the reported " PD-ESR distance measurements are consistent with these predictions". The only conclusion that could be made from these distance measurements for just 1 spin pair is that there is conformational change when lipids are present and that the change brings the labeled sites within the DEER distance range. The rest is just pure speculation. Note that the triangular structure is only based on modelling. It would take more than one distance measurement with more mutants to verify the triangular arrangement experimentally. Please rewrite this section accordingly.

Thank you for the constructive feedback. The DEER data for PLIN3 in solution are indicative of a random polypeptide chain with ~ 100 Å spin-label separation. With liposomes, the protein is structured, and the distance distributions are determined. We do not expect these coarse simulation tools to accurately describe the structure, especially in context of membranes. However, the extent of modeling deviation from the distance constraints is quite reasonable for this stage of the state encouraging further distance mapping work. As we agree for the need to collect more data from distinct pairs to better define the PAT domain structure, we have edited the text in several places to state our conclusions more explicitly; which is that the DEER measurements support a model for the PAT domain to form a folded 3D structure.

Our updated paragraph in the discussion now reads in lines #466-474: *"The PAT domain is predicted to adopt a triangular tertiary structure by both AlphaFold and RoseTTAfold. The DEER distance measurements and CW ESR data are not identical to these predictions, but do indicate that when bound to membranes the PAT domain adopts a folded domain. This conclusion is supported by our HDX-MS results that found the peptides within the PAT domain display longer protection times from H-D exchange, which could be due to either a tertiary structure more resistant to unfolding or a longer membrane residency time. We note that the membrane bound PAT domain structure is likely dynamic and additional distance measurements at distinct sites are needed to verify the accuracy of the predicted triangular structures."*

(3) While the authors discuss the PLIN3 binding to liposomes in terms of membrane lipid packing and associated defects on a qualitative level, there is no discussion of effects of surface charge on binding. Thus, it remains unclear whether the surface charge plays any role in binding. Please discuss the results in terms of what is known about surface charges of the membrane preparations employed in the experiments.

This is an interesting point. We have added some discussion of membrane charge on binding, as well as the polar residue composition of amphipathic helices to PLINs/LD binding.

(4) Several minor issues:

- Please check that all the abbreviations are spelled out in the text.

We have double checked and now believe all abbreviations are spelled out.

- Lines 84-85: "Here, we examined ... and analyze..." – please choose either past or present.

Thanks. We have fixed this issue.

- Lines 294-295: "They (measurements) do not show oscillations, which is frequently observed for membrane proteins. [62-65]". It is not measurements but DEER traces that show oscillations. The latter are typically observed for spin pairs with narrow distance distributions, which are not necessarily related to membrane proteins. Please rewrite.

We apologize for the obtuse sentences. "They" was not meant to speak about "measurements". We have modified this section to read in lines #333-334: "*The decays do not show oscillations, as is frequently observed for spin pairs with narrow distance distributions [68-71].*"

- Lines 306-307: "... recording field-swept echo and did not notice any broadening that could indicate a shorter range of distances (<15 Å)". Note, that the field-swept echo could miss spectral broadening due to short relaxation times of spins at short distances. Continuous wave spectra would be a better measure.

We would definitely see the wings in the field-swept echo, since for MTSL there is no problem with too short relaxation times down to at least 8 Å, where the broad line and low SNR at the micromolar concentrations would make it difficult for us to detect such close pairs. But such close pairs may be present only as a small fraction of spins for the range of MTSL side-chain rotamers expected in lipid environment. We recorded a CW ESR spectrum for 37/114 as suggested. The spectrum only indicated possible presence of distances in the range of 15-20 Å expectedly attenuated in DEER data. Such

distances will not produce conspicuous broadening in the field-swept echo. The distances of 10-15Å would not escape the detection by field-swept primary echo for pulse separation of 250ns. But if there were a minor fraction of such short distances below 10Å, it would be difficult for us to confirm them, given low spin concentration in the sample.

We modified the text accordingly in lines #346-355: *“We also checked the spectral shape by recording field-swept echo with pulse separation of 250ns and did not notice any conspicuous broadening that could indicate a shorter range of distances (<15Å). We do however see from continuous wave (CW) ESR of 37C/114C (Fig. S6) that there might be a sizeable fraction of spins in the 15-20Å range whose contribution to the distance distribution will be significantly attenuated, since DEER has low sensitivity to distances in this range. The conformations with this distance range correlate well with AlphaFold predictions. The spread of the P(r) to longer distances could originate from the mobility of the C-terminal helix where residue 114C is located (Fig. 4C). Taken together, we concluded that the PAT domain does form a folded domain when bound to membranes and this domain is likely mobile with a structure similar but not identical to the AlphaFold and RoseTTAFold predictions.”*

- Lines 280-281: If “PDS is a collection of several ...pulse ESR techniques”, why write “Our PD-ESR distance measurements” in Line 404? Perhaps, “DEER distance measurements”? There is no need for “Our”.

Thank you. We have changed “Our PD-ESR” to “The DEER distance measurements and CW ESR data”

Reviewer #5 (Remarks to the Author):

Overall, the HDX-MS experiments, data, and reporting are of high quality. It is also good to see data have been uploaded to PRIDE. However, there are some issues with the folding/unfolding datasets and the membrane binding datasets having some differences in construction which partially undermines the author’s conclusions. The materials and methods section needs to be generally smartened up also. Comments below:

1) Would the authors comment as to why the 0.3s timepoint is foregone in the membrane binding events? Given that this would provide the best comparison for the membranes facilitating a disorder/order transition.

The 0.3 s timepoint is actually carried out as 3 s of exchange, but performed at 0°C compared to 20°C, which leads to exchange rates that are roughly 10 fold slower. The issue with using this in the membrane binding experiments is that this change in temperature can lead to phase transition effects in the membranes being utilized. This

could lead to different protein binding, and so to simplify interpretation and analysis we always carry out membrane HDX-MS studies at a single temperature.

2) Related to the prior comment – the disorder/order transition experiments were conducted at pH 7, 100 mM NaCl, membrane binding at pH 8, 50 mM NaCl. Why? pH may also drive protein folding, and will of course impact solvent uptake rates, so it adds a layer of possible complexity in the interpretation of disorder/order transition and the membrane binding. Can the authors explain why these differences exist?

Our initial HDX-MS experiments were conducted using pH 7.5 (apologies for the typo, this has been adjusted in the manuscript). In our follow up experiments, we utilized pH 8.0 to match our initial liposome binding assays.

However, to avoid any discrepancy, we have repeated key results of the liposome co-sedimentation assays at different pH and salt concentrations. Please see Figure S2C, which shows membrane binding was not affected by pH or salt concentration.

3) Lines 206-211. There seems to be quite a theoretical leap here with the interpretation of HDX-MS data and membrane residency. Firstly, the authors state two possible scenarios for an explanation of their observed HDX-MS data. However, a third scenario also exists – that the PAT domain/11-mer repeats section remains unfolded and inserts into the membrane i.e. folding and membrane binding may be separate events. Would the authors be able to conduct an additional experiment, such as circular dichromatism, on the 1-196 construct to see if membranes induce alpha-helicity?

We have now conducted CD experiments with full length PLIN3 and constructs for individual regions/domains (Figure S5). The CD spectra are consistent with a shift from random coil in solution to formation of alpha helices in the presence of membranes. The exception to this is the 4-helix bundle region whose CD spectra is unaffected by the presence of liposomes, which we anticipated given this domain is ordered in the absence of membranes.

4) Given the bimodal distributions, has there been an attempt to deconvolute the two populations to determine whether the relative proportions are in keeping with their model of membrane binding?

We did investigate the bimodal populations and provided example spectra in Supplementary figure 3. We made efforts to quantitatively analyze these using bimodal fitting capabilities in our HDX software HDEaminer 2 and HDEaminer 3. Quantitative analysis of the relative populations was extremely noisy, leading to extremely high standard deviation, for this reason we prefer to report on the centroid.

5) There is variation in the description of the pulse experiment as being conducted at either 1oC (Line 167 & 959) or 0oC (Line 498). Please correct as necessary.

Experiments were done at 0°C, we apologize for this typo.

6) Line 499. “20 mM HEPES pH 7” – other values are given to a higher degree of precision. I assume this is 7.0?

This was a typo and was meant to be pH 7.5. Apologies.

7) Line 500 – no volume of quench buffer is provided.

The volume of quench buffer was 60 μ L. We now describe this in the methods

8) Line 499, 509, 519, 511, 520, 521, 533... μ M not uM, the same for volumes. This is repeated throughout the materials and methods section. Additionally, the “ μ ” is occasionally italicised throughout the manuscript. I’m not trying to be a pedant here but it’s all over the place.

We have fixed these typos.

9) Why were different liposome concentrations used between PA and the DAG conditions used?

HDX-MS experiments with 4ME-PC/PE/PA liposomes used a final liposome concentration of 1mM. HDX-MS experiments with DOPC/PE/DAG and 4ME-PC/PE used 0.5 mM. In both cases the lipid concentration should be saturating in protein binding, and this difference in concentration was an oversight between the HDX experiments (performed before and after the COVID lockdown).

10) Line 521 – why was a different pH used in the DOPC liposomes (pH 7.0) cf. pH 8 for PA liposomes used? Is this a typo? It will surely affect the overall exchange kinetics?

This was indeed a typo, all membrane binding experiments utilized pH 8.0. However, membrane binding assays showed no discernible membrane binding difference at either pH 7.0 (Figure. S2C) or 8.0.

11) 539 - 200 °C

We have fixed this typo.

12) No description of how the back-correction for the fully deuterated dataset is provided.

We have described the back correction in more details in the methods.

We added this section in lines #641-644: *“For the experiment with a fully deuterated sample, corrections for back exchange were made by dividing the pulse %D value by the fully deuterated %D value and multiplying by 100. The raw %D incorporation for the fully deuterated sample is included in the source data, with the average back exchange being 33%, and ranging from 10-60%.”*

REVIEWERS' COMMENTS

Reviewer #1 (Remarks to the Author):

The authors have significantly revised their manuscript based on my comments. Hereby they have, for the most part, addressed my concerns. I think this work is now publishable in Nature Communications.

Reviewer #2 (Remarks to the Author):

I think that this work is now suitable for publication.

Reviewer #3 (Remarks to the Author):

All my concerns are adequately addressed and the manuscript is revised accordingly. I do not have new critiques.

Reviewer #4 (Remarks to the Author):

This reviewer is fully satisfied with the revisions made by the authors and would like to recommend the publication of the manuscript in its current form.

Reviewer #5 (Remarks to the Author):

With these additional experiments, the manuscript is much stronger. With the discrepancies between different datasets rectified and CD work especially, the analysis and interpretation are more robust. I am surprised at how noisy the bimodal fitting is, but HDExaminer can be variable in how it interprets bimodal populations.